# Th2 single-cell heterogeneity and clonal distribution at distant sites in helminth-infected mice

Daniel Radtke[1]*, Natalie Thuma[1], Christine Schülein[2], Philipp Kirchner[3], Arif B Ekici[3], Kilian Schober[2], David Voehringer[1]*

[1]Department of Infection Biology, Universitätsklinikum Erlangen and Friedrich-Alexander, University Erlangen-Nürnberg, Erlangen, Germany; [2]Institute of Clinical Microbiology, Immunology and Hygiene, University Hospital Erlangen and Friedrich-Alexander-University Erlangen-Nürnberg, Erlangen, Germany; [3]Institute of Human Genetics, Universitätsklinikum Erlangen and Friedrich-Alexander University Erlangen-Nürnberg, Erlangen, Germany

*For correspondence:
daniel.radtke@uk-erlangen.de (DR);
David.Voehringer@uk-erlangen.de (DV)

Competing interest: The authors declare that no competing interests exist.

**Abstract** Th2 cells provide effector functions in type 2 immune responses to helminths and allergens. Despite knowledge about molecular mechanisms of Th2 cell differentiation, there is little information on Th2 cell heterogeneity and clonal distribution between organs. To address this, we performed combined single-cell transcriptome and T-cell receptor (TCR) clonotype analysis on murine Th2 cells in mesenteric lymph nodes (MLNs) and lung after infection with *Nippostrongylus brasiliensis* (Nb) as a human hookworm infection model. We find organ-specific expression profiles, but also populations with conserved migration or effector/resident memory signatures that unexpectedly cluster with potentially regulatory $Il10^{pos}Foxp3^{neg}$ cells. A substantial MLN subpopulation with an interferon response signature suggests a role for interferon signaling in Th2 differentiation or diversification. Further RNA-inferred developmental directions indicate proliferation as a hub for differentiation decisions. Although the TCR repertoire is highly heterogeneous, we identified expanded clones and CDR3 motifs. Clonal relatedness between distant organs confirmed effective exchange of Th2 effector cells, although locally expanded clones dominated the response. We further cloned an Nb-specific TCR from an expanded clone in the lung effector cluster and describe surface markers that distinguish transcriptionally defined clusters. These results provide insights in Th2 cell subset diversity and clonal relatedness in distant organs.

## Editor's evaluation

CD4 Th2 effector cells contribute to immune responses to helminths and allergens. There exists little information on Th2 cell heterogeneity and clonal distribution between organs. In this manuscript, Radtke et al. investigate the transcriptional signatures of CD4 Th2 cells in the mesenteric lymph nodes and lungs during helminth infection. By using single cell RNA-sequencing including TCR clonotype analysis, the authors define distinct and overlapping transcriptional signatures and clonal relatedness between CD4 Th2 cells in two different tissues at the peak of a type 2 immune response in vivo.

## Introduction

Th2 cells are part of the adaptive immune response against helminths and in allergic diseases. They are recruited and differentiate from a pool of naive CD4 T cells with a wide variety of T-cell receptors

(TCRs) that are formed during T-cell development and provide clonotypic specificity to antigens. Differentiated Th2 cells produce the key type 2 cytokines IL-4, IL-5, and IL-13 that elevate type 2 immune responses and thereby promote allergic inflammation but also mediate protection against helminths (*Walker and McKenzie, 2018*). In recent years, several IL-4 producing Th2 subpopulations have been described and point to substantial heterogeneity within the Th2 population. Only a minor fraction of human IL-4$^+$ T cells produces IL-5 which defines them as highly differentiated cells (*Upadhyaya et al., 2011*). In the mouse, Th2 cells in the lung generally appear more activated and coexpress IL-4 and IL-13 as compared to Th2 cells isolated from lymph nodes of helminth-infected mice (*Liang et al., 2011*). Th2 cells can further differentiate to follicular T helper cells that express IL-4, IL-21, and BCL6 and drive humoral type 2 immune responses in the germinal center (GC) (*Glatman Zaretsky et al., 2009*; *King and Mohrs, 2009*; *Reinhardt et al., 2009*). On the other hand, IL-4 producing T follicular helper (Tfh) cells may develop into strong effector Th2 cells in asthma models (*Ballesteros-Tato et al., 2016*; *Tibbitt et al., 2019*). In addition to IL-4 producing Tfh cells, IL-4 secretion by T cells located outside of GCs can be sufficient for GC formation and class switch recombination to IgE (*Turqueti-Neves et al., 2014*). Tfh13 cells may also develop from Th2 cells in settings or allergic inflammation. These cells coexpress IL-4, IL-5, and IL-13 and promote the generation of high-affinity anaphylactic IgE in response to allergens (*Gowthaman et al., 2019*). In addition to these subsets with distinct functions, there are different activation and developmental stages present in the Th2 population. A rough model describes Th2 cell development as a process that involves activation of naive T cells, followed by proliferation and subsequent production of effector cytokines associated with a high proliferation rate (*Proserpio et al., 2016*). Furthermore, fate mapping and adoptive transfer experiments revealed functional plasticity between T helper cell subpopulations which can lead to Th2 cells with remaining or upcoming signatures of other CD4 T-cell subsets like Th1, Th9, or Th17 cells (*Panzer et al., 2012*; *Peine et al., 2013*; *Tortola et al., 2020*; *Veldhoen et al., 2008*).

Infection of mice with the helminth *Nippostrongylus brasiliensis* (Nb) is a widely used model for human hookworm infections with a strong induction of Th2 responses in lung and small intestine (*Urban et al., 1992*). L3 stage larvae are injected subcutaneously and then first migrate into the lung before they are coughed up, swallowed and reposition to the small intestine where they mature to adult worms. (*Urban et al., 1992*). Using this model, we have previously shown that Nb infection induces a Th2 response with a broad TCR repertoire required for effective worm expulsion (*Seidl et al., 2011*). Development of single-cell sequencing technology now allowed us to gain a deeper understanding of Th2 cell subsets, TCR clonality, and tissue distribution.

Here, we performed single-cell sequencing of TCR genes combined with transcriptome profiling of Th2 cells isolated from mesenteric lymph nodes (MLNs) and lung of IL-4eGFP reporter mice (4get mice) at day 10 after Nb infection. In this manuscript, we use the term 'Th2' for the CD4$^+$, IL-4eGFP$^+$ cells. This likely includes cells licensed for IL-4 expression that will not develop into terminally differentiated Th2 cells and retain functional plasticity, and NKT cells. By our approach, we revealed heterogeneity and differentiation paths within the Th2 compartment, compared Th2 population similarity at distant sites, analyzed cell exchange between organs by clonal relatedness and characterized expanded clones and their TCR sequences.

## Results

### Th2 cells show an organ-specific gene expression profile consistent with acquired effector functions

We performed combined transcriptome and TCR clonotype analysis using the chromium 10× Genomics and Illumina single-cell RNA sequencing (scRNAseq) platform on IL-4-expressing Th2 cells isolated from perfused lung (also containing a remaining fraction of intravascular cells) and MLN of two IL-4eGFP reporter (4get) mice (*Mohrs et al., 2001*) that had been infected 10 days before with Nb (*Figure 1A*). IL-4-expressing Th2 cells (CD4$^+$IL-4eGFP$^+$) were sorted from single-cell suspensions of both organs and were directly subjected to scRNA library preparation.

4get mice were chosen as they allow isolation of Th2 cells ex vivo without prior restimulation. In contrast to other IL-4 reporter mice such as the KN2 strain, 4get mice even report the early stages of Th2 differentiation (*Mohrs et al., 2005*). Sampling of the lung was performed as Th2 cells accumulate in this organ a few days after Nb infection. Complimentary MLNs were included as a distant secondary

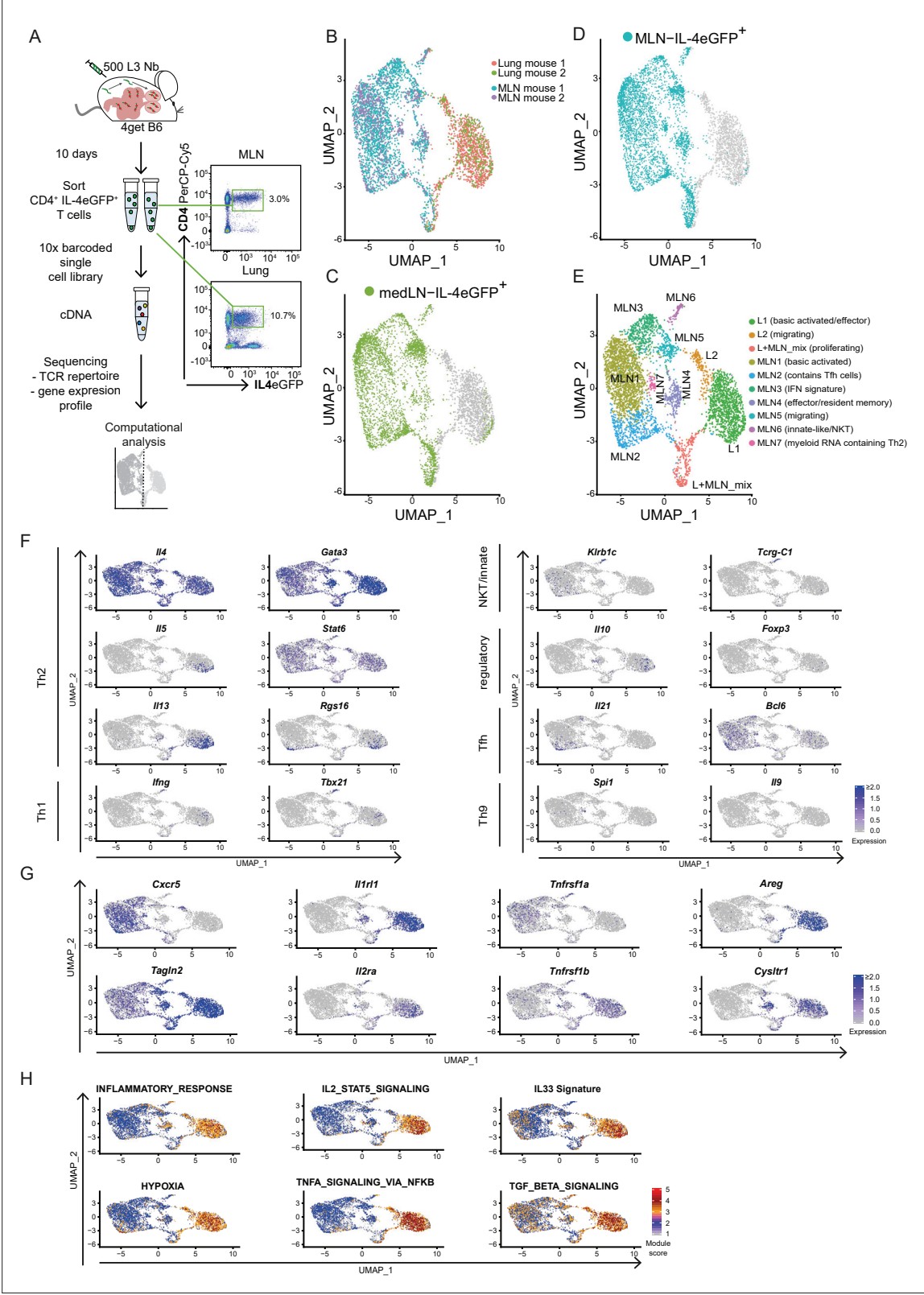

**Figure 1.** Th2 cells of mesenteric lymph node (MLN) and lung adopt tissue-specific RNA signatures. (**A**) General experimental outline. MLN and lung cells of two individual *Nippostrongylus brasiliensis* (Nb)-infected IL-4eGFP reporter mice (4get B6) were sorted for IL-4eGFP+CD4+ cells 10 days post infection. Then combined transcriptome and T-cell receptor (TCR) repertoire sequencing were performed. Flow cytometry plots show the frequency of Th2 cells (IL-4eGFP+CD4+ cells) in MLN and lung. (**B**) Uniform Manifold Approximation and Projection (UMAP) representation of MLN and lung cells

*Figure 1 continued on next page*

*Figure 1 continued*

10 days post Nb infection. (**C, D**) medLN and MLN cells sequenced in a separate run were plotted on the existing UMAP defined in the initial MLN + lung sequencing run by integration based on shared anchor genes to demonstrate similarity of medLN and MLN cells. (**E**) De novo unsupervised clustering approach with manually added cell type description. Clusters are indicated on UMAP. (**F**) Expression of selected CD4 T-cell subset defining genes, (**G**) genes that are differently expressed between MLN and lung, or (**H**) gene signature module scores for single cells plotted on top of UMAP representation. Each of the independent single-cell sequencing experiments is based on two mice.

The online version of this article includes the following figure supplement(s) for figure 1:

**Figure supplement 1.** Quality control of Th2 single-cell sequencing 10 days post *Nippostrongylus brasiliensis* (Nb) infection.

**Figure supplement 2.** Comparison of mesenteric lymph node (MLN) and lung Th2 cells after *Nippostrongylus brasiliensis* (Nb) infection on single-cell level.

**Figure supplement 3.** Comparison of medLN and mesenteric lymph node (MLN) Th2 cells after *Nippostrongylus brasiliensis* (Nb) infection on single-cell level.

lymphoid organ associated to the intestine where the worms reside approximately from days 4 to 10 after infection. By choosing a secondary lymphoid organ and a peripheral organ we cover location-specific differences as well as a broad spectrum of developmental stages. In both organs worms of larval stage 4 are present in the course of the infection (*Camberis et al., 2003*) and provide a common antigen basis in addition to antigens shared between larval stages or systemic dissemination of antigens from larval secretions or dead larvae. This setup enabled us to compare Th2 cell subsets and clonotypes in both organs at single-cell resolution.

In order to restrict the analysis after sequencing to high-quality Th2 cells, we included in total 4710 cells with detected, functional TCR α- and β-chains that also passed our additional QC filters (see Materials and methods) (*Figure 1—figure supplement 1*). We used an unbiased high-dimensional clustering approach followed by dimensional reduction for simple representation of complex data (*Stuart et al., 2019*).

Our approach revealed that Th2 cells of lung and MLN have a distinct organ-specific expression profile represented by clear separation of cells from both organs upon dimensional reduction (*Figure 1B*) and highlighted by differential expression analysis between lung and MLN cells (*Figure 1—figure supplement 2A*). We mainly focused on the analysis of MLN and lung in order to compare the T-cell repertoire between distant organs later on. Importantly, Th2 cells from lung draining mediastinal LN (medLN) and MLN analyzed in a separate scRNAseq experiment (*Figure 1—figure supplement 3*) showed a similar expression profile when layered on the lung–MLN UMAP of our initial experiment (*Figure 1C,D*). This indicates that most of the differences are due to the comparison of a secondary lymphoid organ with a peripheral organ, and only minor differences are related to the comparison of distant sites (medLN vs. MLN).

We continued with unbiased clustering of MLN and lung to identify two lung, seven MLN, and a mixed proliferative cluster that are numbered by size. These are summarized here and further described in the following two sections (*Figure 1E*): L1 (basic activated/effector), L2 (migrating), L + MLN (proliferating), MLN1 (basic activated), MLN2 (contains Tfh cells), MLN3 (IFN response signature), MLN4 (effector/resident memory like), MLN5 (migrating), MLN6 (innate-like/NKT), and MLN7 (myeloid RNA containing Th2). For every cluster, the 10 marker genes that are upregulated in a cluster compared to all other cells in the dataset were determined as a reference (*Figure 1—figure supplement 2B*).

We first concentrated on markers for known CD4 subpopulations. As expected, the cells from MLN and lung express the Th2 hallmark genes *Il4*, *Gata3*, and *Stat6* (*Figure 1F*). However, almost exclusively the L1 cluster expresses IL-5 which promotes eosinophil development, recruitment, and survival. Similarly, IL-13 is expressed a bit broader in L1 but additionally in lymph node cluster MLN4. IL-13 elicits a broad spectrum of effector type 2 immune functions including eosinophilic inflammation, mucus secretion, and airway hyperresponsiveness (*Rothenberg and Hogan, 2006*; *Takatsu et al., 2009*). According to the proinflammatory IL-5 and IL-13 production, double producers are thought to be a strong or pathogenic effector subset of Th2 cells that includes highly differentiated CD27[low], PD-1(*Pdcd1*)[high] memory cells (*Upadhyaya et al., 2011*), which is also reflected on gene expression level in our data. Enhanced *Rgs16* expression of the IL-5[+]/IL-13[+] cells is also associated with higher cytokine production (*Lippert et al., 2003*) and further supports effective effector molecule production (*Figure 1F*).

As expected, very few IL-4-positive cells express the Th1 hallmark genes *Ifng* and *Tbx21* (encodes T-bet). However, cells that express IL-4 and both markers are present in cluster MLN6. An overlapping fraction of cells also expresses *Zbtb16* which encodes the NKT cell-associated transcription factor PLZF (*Savage et al., 2008*) and *Klrb1c* (encodes NK1.1) or in addition to the TCRα and TCRβ chains *Tcrg-1c* (encodes the constant region of TCRγ) as an indication that these cells show signs of unsuccessful or not yet successful development into γδ T cells. Hence, MLN6 seems to contain predominantly innate-like/NKT cells (*Figure 1F*).

The classical marker for Treg cells FOXP3 was hardly found on gene expression level in our dataset. Nevertheless, small fractions of cells in L1 and MLN4 express *Il10*, which suggests regulatory capacity independent of *Foxp3* expression (*Figure 1F*) that has also been reported as relevant to maintain the Th2 response and to repress Th1 (*Balic et al., 2006*). Interestingly, the potential regulatory IL-10 producers cluster together with the likely proinflammatory IL-5 and IL-13 producers, suggesting a similar expression profile of cells with regulatory and effector function.

Tfh cells that express IL-4 were detected in MLN2. They express both Tfh markers IL-21 and BCL6 on gene level. Of note, *Bcl6* expression seems less restricted to a specific cluster. Where *Il21* and *Bcl6* expression overlaps cells show expression of *Rgs16* associated with enhanced Th2 cytokine production and trafficking (*Lippert et al., 2003*). In line with a recent publication, we did not observe Tfh13 cells (IL-13^hiIL-4^hiIL-5^hiIL-21^lo) which were reported to be associated with production of high-affinity anaphylactic IgE in Th2 responses to allergens but not helminth infections (*Gowthaman et al., 2019*).

Increased expression of TGFβ-associated genes further suggested Th2 cell plasticity toward Th9 cells in our dataset (*Veldhoen et al., 2008*; *Figure 1H*) and IL-9 is described as a relevant factor for hookworm expulsion (*Licona-Limón et al., 2013*). However, we only find very few *Il9* or *Spi1* (encodes the Th9-associated transcription factor PU.1) expressing cells in the lung. *Il9* expression was also barely detectable in the MLN, while *Spi1* was expressed in the MLN4 population potentially describing an early stage of Th9 development (*Figure 1F*).

Next, we screened for genes that distinguish Th2 cells from MLN and lung and that might reflect organ-specific differences in gene expression or Th2 development. While most cells from the MLN (MLN1–3) express the gene for the chemokine receptor CXCR5 associated with homing to B cell follicles and recruitment of Tfh cells to GCs (*Breitfeld et al., 2000*; *Schaerli et al., 2000*), the majority of Th2 cells from the lung and MLN cells that cluster in proximity to lung cells (MLN4–6) hardly express it (*Figure 1G*). In contrast, the lung and MLN4–6 cluster cells express high levels of the gene for TAGLN2 which stabilizes the immunological synapse and is relevant for proper T-cell effector function (*Na et al., 2015*). A stronger effector phenotype of lung cells is also supported by an increase of inflammation signature genes and hypoxia-associated genes in these cells, which are associated with enhanced glycolysis required for late Th2 effector differentiation (*Healey et al., 2021*; *Stark et al., 2019*; *Figure 1H*).

In line with strong effector function, most lung cells and some nearby projected MLN cells express the gene for the IL-33 receptor ST2 (*Il1rl1*) (*Figure 1G*). It recognizes the alarmin IL-33 and induces production of key type 2 cytokines IL-5 and IL-13. This is crucial for the clearance of intestinal helminths as mice that lack IL-33 are not able to effectively cope with the infection, likely due to defects in the T cell and ILC2 compartments (*Hung et al., 2013*). In line with enhanced *Il1rl1* expression, we find evidence that ST2 inducing pathways are active in lung cells. As such, there is higher expression of IL-2–STAT5 axis target genes and the IL-2 receptor alpha-encoding gene itself (*Il2ra*) (*Guo et al., 2009*; *Meisel et al., 2001*) as well as enhanced tumor necrosis factor receptor 2 (TNF-R2) transcript expression and correspondingly an increased TNF induced expression signature, suggesting stimulation of the cells by TNF (*Kumar et al., 1997*; *Figure 1GH*). Looking further downstream, we find an elevated gene signature for IL-33-stimulated T cells in the lung, which suggests active signaling via the ST2 receptor, associated with pathogen clearance. However, IL-33/ST2 not only invokes effector mechanisms against the worm but also promotes production of *Areg*, encoding amphiregulin, that we find upregulated and which is involved in tissue repair and resolution of inflammation after Nb infection (*Minutti et al., 2019*; *Figure 1G*). However, amphiregulin can also promote reprogramming of eosinophils to develop fibrosis in other setups (*Morimoto et al., 2018*) with potential implications on longer-term pathology after Nb infection (*Marsland et al., 2008*).

In our Nb infection model, lung cells and a fraction of proximal clustering MLN cells also express genes associated with asthma or involved in pathways targeted by drugs for asthma treatment like

*Cysltr1*, *Plac8*, or *Adam8* (*Naus et al., 2010*; *Tibbitt et al., 2019*; *Trinh et al., 2019*). Lung cells but only few MLN cells in our dataset also show an increased expression of TGFβ target genes, consistent with the described Th2 cell plasticity toward a Th9 phenotype (*Veldhoen et al., 2008*) which potentially further broadens the T effector functions (*Figure 1F–H*).

## Conserved expression profiles for migratory and effector/resident memory Th2 cell populations in lung and MLNs

As a next step after concentrating on known CD4 subpopulations and comparison of MLN and lung on organ level, we focused on similarities of MLN and lung and clusters with unexpected signatures.

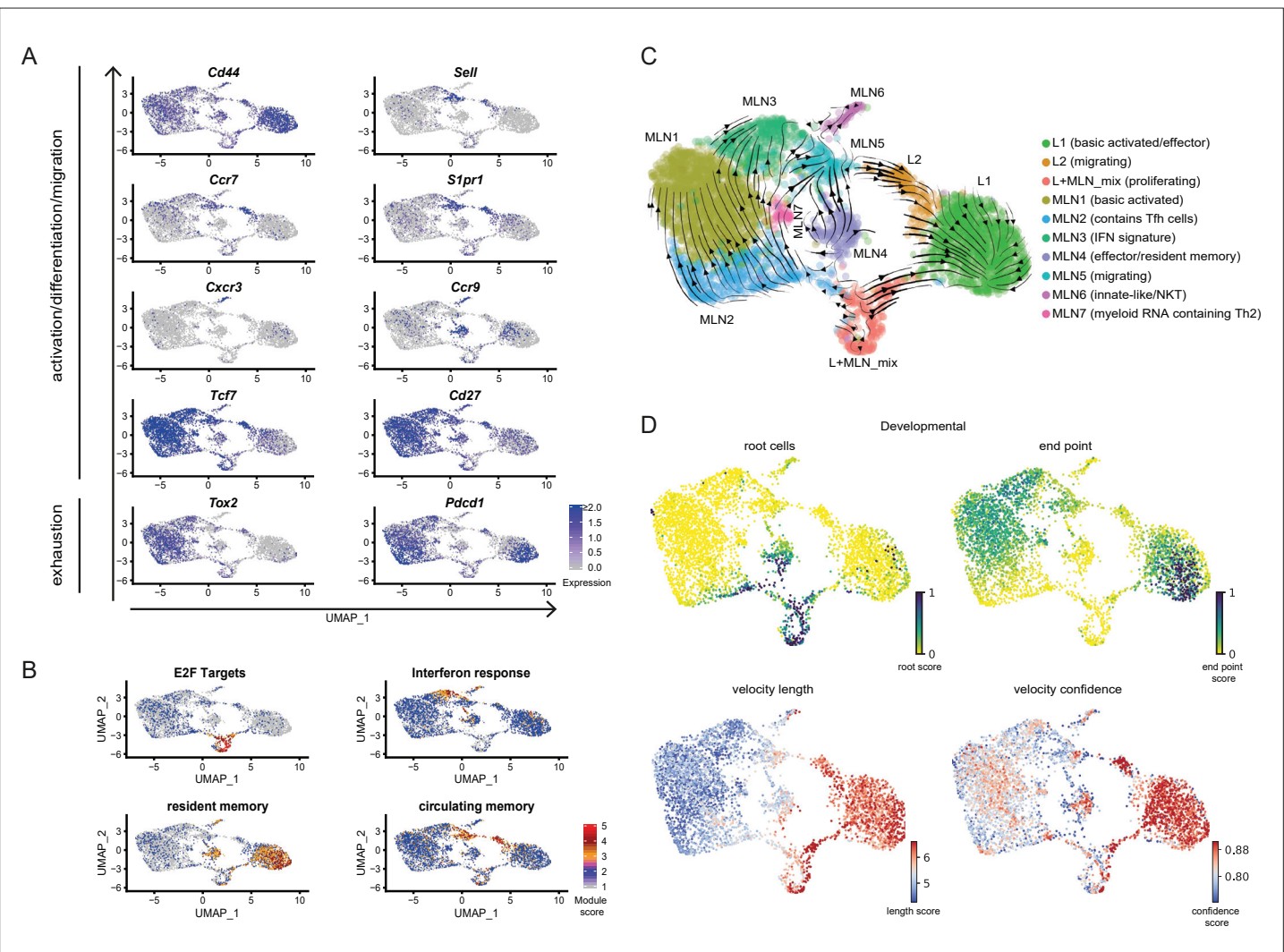

**Figure 2.** Conserved expression profiles for migratory and effector Th2 populations between organs and their inferred developmental paths. (**A**) Expression of selected marker genes associated with biological processes or (**B**) gene signature scores for single cells on top of UMAPs. (**C**) RNA velocity analysis. Arrows present inferred developmental paths. (**D**) RNA velocity defined root cells and developmental endpoints as well as inferred differentiation speed (velocity length) and velocity confidence for cells are visualized on UMAPs.

The online version of this article includes the following figure supplement(s) for figure 2:

**Figure supplement 1.** Vasculature stain for IL-4eGFP +CD4+ subsets from lung tissue.

**Figure supplement 2.** Integrative similarity analyses of mesenteric lymph node (MLN) and lung Th2 cells after *Nippostrongylus brasiliensis* (Nb) infection.

**Figure supplement 3.** Prediction of developmental paths for Th2 cells of mesenteric lymph node (MLN) and lung after *Nippostrongylus brasiliensis* (Nb) infection.

Cluster L2 and MLN5 visually form a 'bridge' between the MLN and lung compartments. Indeed cells in these clusters both express genes coding for CD62L (*Sell*), CCR7, and S1PR1 involved in cell adhesion and T-cell trafficking suggesting that these clusters contain recent immi-/emmigrants (*Figure 1E* and *Figure 2A*). Of note, staining of cells in the blood circulation revealed that a fraction of cells in the L2 cluster likely represents cells located in blood vessels with a gene expression profile similar to tissue-resident L2 cluster cells which further indicates that the L2 cells might come from or migrate into blood vessels (*Figure 2—figure supplement 1*). The 'bridge clusters' also express *Tcf7* (associated with self-renewal capacity) and *Cd27* (encoding a central memory T-cell marker) (*Figure 2A*). In line, the whole 'bridge' shows a circulating memory signature (*Rahimi et al., 2020*; *Figure 2B*). L2 part of the 'bridge' is the only lung fraction that expresses reasonable amounts of *Cxcr5* and *Tox2*, which are expressed in a majority of cells in the MLN (*Figures 1G and 2A*). This suggests that the profile of these lung cells in part reflects the profile found in secondary lymphoid organs and strengthens their identification as migrating cells. There are also differences between the lung- and MLN-associated 'bridge' clusters. Cells in L2 expressed more CD44, which is suggestive of cells in later central memory or effector cell state and exhaustion marker genes encoding *Tox2* and *Pdcd1* compared to MLN5. In contrast, to L2 which shows signs of MLN-associated gene expression, MLN5 does not show clear signs of a lung signature (*Figure 1F, H* and *Figure 2A*), potentially reflecting that the visual 'bridge' is not a real connection and contains immi-/emmigrants to/from other secondary lymphatic organs like the lung draining lymph nodes or other peripheral organs.

Proliferating cells of both organs have a similar expression profile and fall into the same cluster (L + MLN) as proliferation induces a strong gene signature highlighted by a proliferation-associated E2F signature gene set (*Figure 2B*).

We find three effector clusters in our dataset, L1 lung cells express effector-associated genes (e.g., stronger expression of *Il1rl1*, *Cysltr1* [receptor for cysteinyl-leukotrienes C$_4$, D$_4$, and E$_4$], *Il2ra*, *Il5*, and *Il13*) but MLN4 has a similar signature. Both populations are high for a published signature of lung-resident memory T cells (*Rahimi et al., 2020*; *Figure 2B*) and for both the effector-associated genes *Plac8* and *Adam8* drive the signature (*Figure 1—figure supplement 2B* and not shown). In contrast to L1, most MLN4-resident memory signature cells express the gene-encoding CCR9 relevant for lamina propria homing (*Campbell and Butcher, 2002*; *Stenstad et al., 2006*), while lung cells express the gene for CCR8 hardly found in the MLN (not shown). In addition, to MLN4 there is an innate-like/NKT cluster (MLN6) present in the MLN that also shows expression of effector genes. Both clusters express a resident memory signature and *Ccr9*, probably reflecting local effector/resident memory populations that participate in the intestinal immune response. Of note, there is also a *Ccr9*-expressing fraction of cells in L1 with little overlap to the *Il5/Il13* secreting cells (*Figure 2A*). These cells, positioned in the lung might come from or might migrate to the lamina propria. Interestingly, the innate/NKT MLN6 effector cluster expresses the gene for the usually Th1-associated chemokine receptor CXCR3 found before (*Kim et al., 2001*) in a fraction of Th2 cells but they were not linked to innate/NKT phenotypes. Therefore, there is an expected effector cluster in the lung but cells with an effector signature are also found in the MLN which is mainly recognized to contain earlier stages in T helper cell development. Despite the shared effector characteristic of effector-associated clusters, there is site-specific gene expression likely reflecting local effector function. Similarly, there is also evidence for site-specific effector signatures in our second sequencing run comparing medLN and MLN. medLN contains a number of cells that rather cluster with lung effector cells (L1) and a reduced number of cells that show a lamina propria associated effector signature (like MLN4) compared to MLN (*Figure 1C,D*).

To highlight the similarities of 'brigde' (MLN5, L2) and effector clusters (L1, MLN4) between MLN and lung, we integrated the lung and the MLN data computationally based on shared expression anchors before performing dimensional reduction (*Stuart et al., 2019*). The approach reveals that bridge clusters as well as effector clusters (L1 and MLN4) show the most similar expression profiles between MLN and lung and hence cluster together in low-dimensional space projection (*Figure 2—figure supplement 2*).

While Th1 and Th2 cells are often seen as counterparts that can antagonize differentiation of each other, there is also evidence that the Th1 hallmark cytokine IFN-γ promotes proper Th2 heterogeneity and differentiation either directly as suggested by in vitro differentiation studies (*Wensky et al., 2001*) or indirectly by inducing activation of DCs with Th2-priming capacity (*Connor et al., 2017*; *Webb*

*et al., 2017*). In accordance MLN3 expresses an IFN response signature after Nb infection (*Figure 2B*) likely associated with Th2 priming and differentiation.

We further identified an unusual subset of cells (MLN7) which contains genes for the MHCII-associated invariant chain (CD74), complement component C1q, lysozyme, and CD209b. Some of these genes are rather associated with myeloid cells like DCs or macrophages. This population might therefore represent cells emulsified with exosomes or RNA containing vesicles during library generation, either externally attached or taken up by the cell.

In summary, our dataset with high resolution for the Th2 compartment after Nb infection reveals distinct subsets of Th2 cells in lung and MLN including a 'bridging' cluster of cells with closely related gene expression profiles in both organs. Defined clusters of Th2 subsets were later used to link TCR clonotypes to specific clusters.

## RNA-inferred developmental directionality of Th2 cells supports proliferation as a hub for differentiation decisions

To infer developmental relatedness of clusters defined above we performed an RNA velocity analysis in which the ratio of spliced to unspliced RNA transcripts is used to calculate and visualize likely developmental directions (*Bergen et al., 2020*; *La Manno et al., 2018*; *Figure 2C*). We used the scVelo algorithm (*Bergen et al., 2020*) which identifies the most undifferentiated cells as root cells and highly differentiated cells as developmental endpoints, connected by arrows that show likely paths from root to endpoints (*Figure 2C,D*). The proliferation cluster (L + MLN) reflects the majority of root cells in our data and highlights proliferation, in accordance with the literature (*Gett and Hodgkin, 1998*; *Gulati et al., 2020*; *Radtke and Bannard, 2018*), as a critical branching point at which differentiation decisions are taken. The developmental direction going from proliferating cells toward the other populations is also supported when RNA velocity is performed on PCA level (*Figure 2— figure supplement 3A*), in paga projection (*Figure 2—figure supplement 3B*), in the static model of RNA velocity (*Figure 2—figure supplement 3C*), in developmental analysis performed with the monocle package (*Trapnell et al., 2014*; *Figure 2—figure supplement 3D*) or when cell cycle was regressed out before RNA velocity analysis (*Figure 2—figure supplement 3E*). Another algorithm that is described to robustly detect developmental potential also for proliferating cells additionally indicates that the proliferating cluster has the highest developmental potential (*Gulati et al., 2020*; *Figure 2—figure supplement 3F*).

The MLN part of the 'bridge' clusters (MLN5), the IFN signature cluster (MLN3), and the main MLN (Basic activated Th2; MLN1) cluster are marked as relatively diffuse endpoints in the MLN, associated with a low differentiation speed and confidence reported by scVelo (*Figure 2D*). This suggests that wide parts of the MLN Th2 cells are relatively heterogeneous. The effector/resident memory like cluster (MLN4) is in itself heterogeneous and contains cells with a strong root signature which hardly overlap with the also contained strong resident memory signature cells. A relatively high differentiation speed and confidence compared to other MLN clusters suggest that it contains a fast developing effector/resident memory like population. Based on the MLN5 'bridge' cluster definition as an endpoint, MLN5 might rather reflect cells that leave the MLN. The lung cluster of the 'bridge' (L2) instead contains cells that differentiate with high confidence and inferred speed toward the main lung cluster of effector cells (L1), which suggests that these cells enter the lung and further differentiate locally. The IL-5/IL-13 double producers previously defined as highly differentiated effector cells (*Upadhyaya et al., 2011*) reflect the endpoint in the lung (*Figure 2C and D*).

In conclusion, RNA velocity supports proliferation as a hub for differentiation in the Th2 compartment and supports that migratory Th2 cells rather leave secondary lymphatic organs and enter peripheral organs while the reverse migration path is a rare event.

## Longitudinal study of Th2 development on protein level based on a transcriptionally defined markers

In an attempt to relate described transcriptional profiles and cluster definitions to distinguishable surface receptor signatures, we analyzed expression of identified surface marker genes on the protein level by flow cytometry. We included MLN and lung but also medLN to reveal differences in Th2 development at different sites over time. First, we selected differentially expressed surface markers that allow to distinguish defined clusters and analyzed their expression on Th2 cells isolated from

Nb-infected 4get mice longitudinally (d0, d6, d8, and d10) (*Figure 3*). CD4 cells of all samples were computationally gated, extracted, and concatenated before downsampling and dimensional reduction (UMAP) were applied based on Cxcr6, CD62L, Cxcr5, CD279 (PD-1), Ly6a/e, Itgb7, CD127, CD74, CD44, CD4, and CD3e expression, but not on IL-4eGFP expression to resolve similarities between IL-4eGFP⁺ and IL-4eGFP⁻ cells. Staining pattern for single markers (*Figure 3—figure supplement 1A*) and definition of gates aimed to resemble transcriptionally defined populations on protein level (*Figure 3—figure supplement 1B*) are given. Data of time points d6, d8, and d10 splitted by organ are shown either for all CD4⁺ cells or only for IL-4eGFP⁺CD4⁺ cells (*Figure 3B*). IL-4eGFP⁺CD4⁺ cells of the lung show a different distribution over the UMAP than LN cells, with the majority of cells falling into the L1 gate while the L2 population is less well defined by chosen markers. There is also additional heterogeneity with a number of cells also present in gates defined for MLN cells. As on transcriptional level, the distribution of MLN cells seems again comparable between medLN and MLN. Next, concatenated CD4⁺ T cells (*Figure 3—figure supplement 1C*) or only IL-4eGFP⁺CD4⁺ T cells (*Figure 3C*) were splitted based on organ and time after infection. In the lung, the L1 effector cluster increases with infection time in accordance with a developing effector response against Nb. Cells in the L2 gate were rather variable and there is no clear trend. The medLN response seems a bit faster than the MLN response but overall similar. Effector cells (in L1, MLN4, and MLN6 gate) moderately increase over time while population MLN2 is increased and MLN3 is strongly increased at d6 and d8 after Nb infection. This pattern is maintained at d10 for MLN but medLN population MLN1, MLN2, and MLN3 are all expressed to the same extend. It might therefore be that MLN3 cells are precursors of MLN2 and MLN1 as we see the proportional increase later on. That the increase in MLN1 and MLN2 signature cells at d10 is not seen in MLN might reflect a delayed development related to the delayed occurrence of worms in the intestine compared to the lung and medLN.

In summary, we established a surface staining strategy that allows to track Th2 subpopulations during Nb infection by flow cytometry.

## Clonal relatedness of Th2 cells in distant organs confirms effective exchange of effector cells

The single-cell immune profiling approach allows for combined RNA expression profiling and TCR repertoire analysis, which made exploration of clonal relatedness between clusters and efficient distributed of clones across organs possible. This analysis also aimed to define potent Nb-specific TCRs based on the hypothesis that these might belong to successfully expanded clones associated with above defined effector clusters.

In line with previous results (*Seidl et al., 2011*), we find a broad TCR repertoire after Nb infection as the majority of distinct TCRs was only found in one cell. However, 28% of cells expressed a TCR found in at least two different cells (same CDR3 nucleotide sequence, the same variable and joining segments). The most abundant clone has 15 sequenced members in the samples analyzed which translates to about 8600 estimated members in total lung and MLN tissue. As we only analyze a small sample of the whole organs the calculation clearly underestimates the fraction of expanded T-cell clones in the population. A still substantial part of clones is found in both organs, which suggests an effective distribution between MLN and lung. In contrast, only two small clones were identical between the two analyzed mice implicating very few public clones (*Figure 4A*). The innate-like/NKT innate cluster (MLN6) and the MLN7 cluster hardly contained expanded clones suggesting limited TCR specificity-driven proliferation in these clusters (*Figure 4B*). Cells of the 'bridge' clusters (MLN5 and L2) contain substantially more expanded clones but less than the effector/resident memory like populations (MLN4 and L1), which in turn contain less expanded clones than the more homogeneous majority of MLN clusters (MLN1, MLN2, and MLN3). It might reflect that the 'bridge' clusters contain immi-/emmigrated cells from distant sites with less clonal overlap to the local population.

Next, we find that strongly expanded clones are effectively spread over organs (*Figure 4C*). The typical caveats of current single-cell technologies (sampling noise and limited sample size) do not allow to draw a similar conclusion for lowly expanded clones (<3 cells per organ). The determination of the clonal relatedness of clusters compared to the overall frequency of a cluster in the dataset again highlights effective distribution of effector Th2 cells between distant organs (*Figure 4D*). The clones of clusters that are most distant to cells of the other organ (L1, MLN1, MLN2, and MLN3) tend to expand more locally, represented by the higher percentage of related cells found in the same organ compared

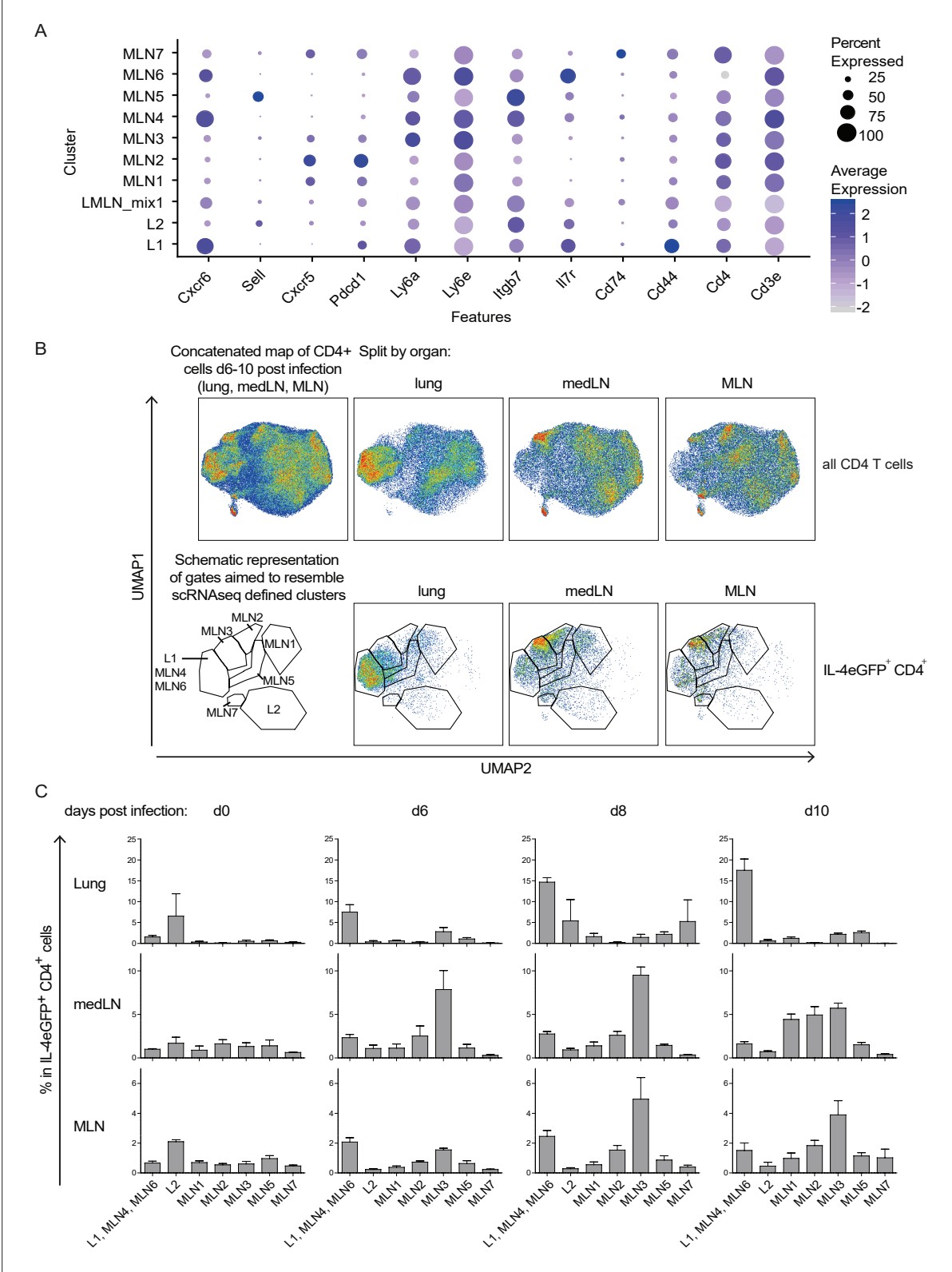

**Figure 3.** Th2 subset distribution in *Nippostrongylus brasiliensis* (Nb) infection time course. CD4-expressing cells from lung, medLN, and mesenteric lymph node (MLN) of IL-4eGFP reporter (4get) mice were stained for markers defined by single-cell sequencing at indicated days after Nb infection. (**A**) Selection of transcriptionally defined marker genes encoding for surface receptors. (**B**) Surface staining for selected markers on protein level was performed followed by flow cytometric analysis. CD4+ cells were computationally selected, down sampled and a UMAP of the data was generated.

*Figure 3 continued on next page*

*Figure 3 continued*

Shown are CD4 T cells of all organs after infection (days 6, 8, and 10 post Nb infection) concatenated into one UMAP and splitted by organ (upper row). The same is given for IL-4eGFP⁺CD4⁺ cells (lower row). Gates are aimed to resemble transcriptionally defined cluster on protein level. (**C**) Quantified distribution of IL-4eGFP⁺CD4⁺ cells in UMAP gates over the course of Nb infection. For (B, C), *n* = 3 mice per time point; error bars represent standard error (SE).

The online version of this article includes the following figure supplement(s) for figure 3:

**Figure supplement 1.** Surface receptor stain aimed to resemble transcriptionally defined clusters on protein level.

to the overall distribution of clusters. Directly compared to those clusters, the clones in 'bridge' clusters (L2 and MLN5) have a higher frequency of members in the other organ, especially apparent in the other part of the 'bridge' in each case. The effector/resident memory like cluster of the MLN (MLN4) also shows increased relatedness to the lung that contains a large number of effector cells in cluster L1. The finding that clusters visualized near the other organ also show enhanced TCR repertoire relatedness to that organ confirms significant migration between organs and that the UMAP efficiently displays real relatedness of clusters.

As a next step, we visualized the five most strongly expanded clones determined for each organ separately or we combined counts to get the most strongly expanded clones in the total dataset (*Figure 4E*). Members of such clones in the MLN tend to be preferentially found in the MLN1 cluster and to a lower extend in the neighboring Tfh-associated cluster (MLN2) and the IFN signature cluster (MLN3). They are hardly found in the lung proximal clusters (MLN4–MLN7). However, few members of four of the top expanded MLN clones are found in the lung and are therefore successfully spread across organs. Top expanded lung clones do not overlap with the top expanded MLN clones and preferentially show up in the big lung cluster in which effector cells are found (L1). In contrast to the MLN only one of the top expanded lung clones has members in the MLN, indicating that these clones successfully expanded locally but have limited capacity to spread to the MLN. The five most highly expanded clones in the whole dataset strongly overlap with the ones determined for the separate organs. This indicates that despite remarkable exchange between the distant organs, strong local expanders dominate the response and while more evenly distributed clones are present, they do not outnumber locally expanded ones in a combined analysis of MLN and lung cells.

In summary, there is substantial overlap of expanded clones between the MLN and lung during Nb infection, but rather locally restricted clones successfully expanded in an otherwise diverse pool of Th2 cells.

## No general preference for specific TCR chain compositions

After analysis of single clones in the last part, where we found expansion but no obvious dominant clones, we determined if there are rather preferentially used TCR segments or segment combinations that could suggest potent effector function of TCRs against Nb. First, we included only one representative member per clone and compared if the same combination of TCRα and TCRβ chain segments is shared between the top 50 most frequently used segment combinations in both organs, both analyzed mice and in nonexpanded versus expanded clones (*Figure 5A*). For the nonexpanded clones there was hardly any overlap seen between mice or organs, only two of the fifty combinations were found in three of the four analyzed organs (MLN and lung of two mice). For the expanded clones, there was limited overlap in combined segment usage between organs of one mouse but not the other. Combinations of TCRα- or TCRβ variable with joining segments and TCRα with TCRβ variable segments also revealed limited overlap in the top used combinations. *Trbv1* was a recurrently used TCRβ variable segment present in frequently used combinations (*Figure 5—figure supplement 1A*). Similarly, for single segments there was no obvious preferential usage in expanded clones compared to nonexpanded ones. Again, *Trbv1* was one of few constituents that was moderately increased in expanded versus nonexpanded clones (*Figure 5—figure supplement 1B,E*). In addition, there was no striking difference observed in total CDR3 amino acid length/length distribution that could be indicative for changes in specificity (*Davis et al., 1998*; *Rock et al., 1994*) between expanded and nonexpanded clones (*Figure 5—figure supplement 2A*). In a finer grained analysis of single TCRα and TCRβ family members, there was also no change in CDR3 length or length distribution (*Figure 5—figure supplement 2B,C*). The general TCRα or TCRβ CDR3 length in MLN and lung of the Nb-infected mice is also not altered compared to naive T cells of the peripheral blood (*Figure 5B*).

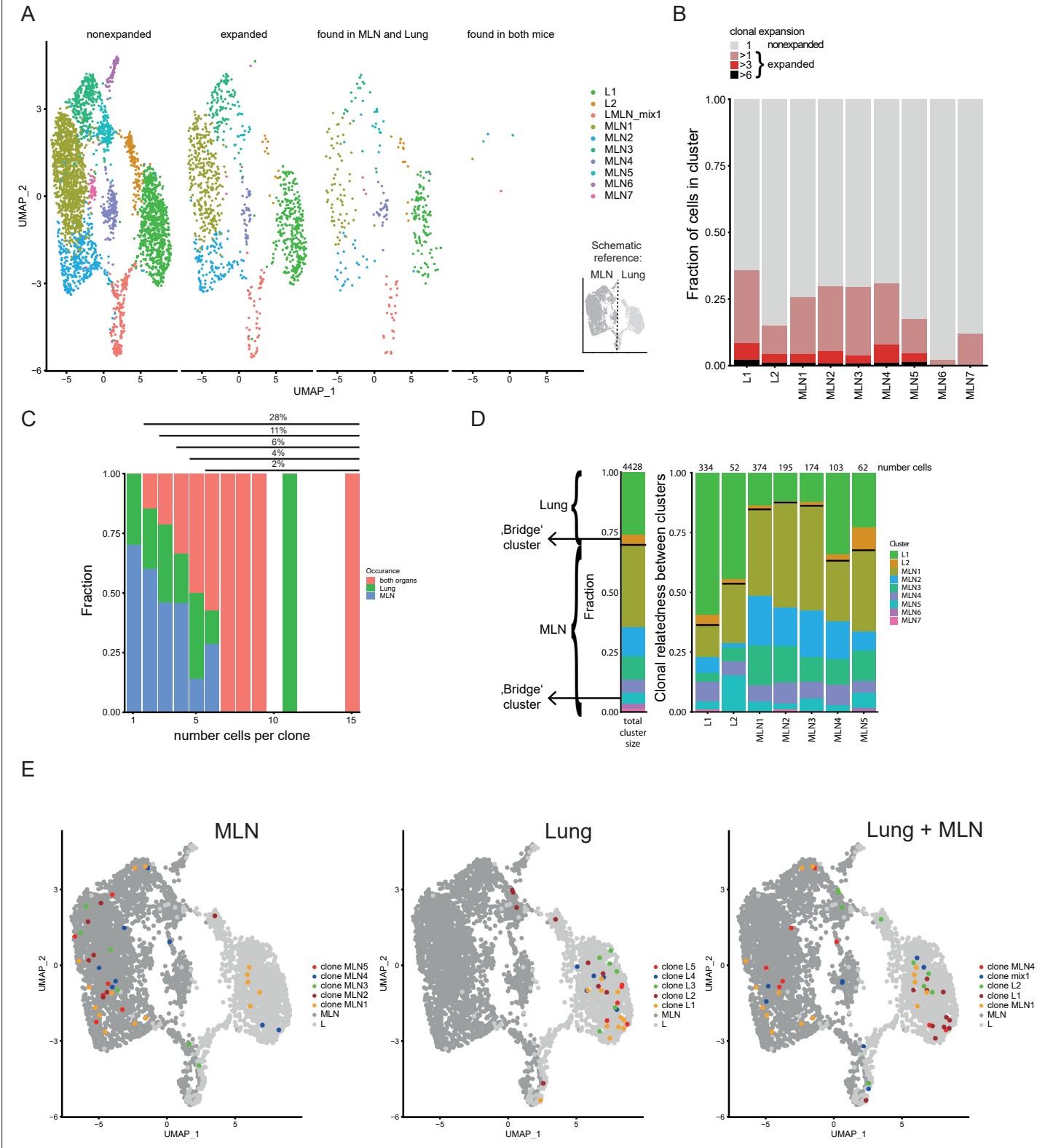

**Figure 4.** Clonal relatedness of Th2 cells in mesenteric lymph node (MLN) and lung. (**A**) UMAP of MLN and lung cells split by cells with unique T-cell receptors (TCRs) (nonexpanded), cells with the same TCRs found in more than one cell (expanded), cells with the same TCRs found in both organs and cells with the same TCRs found in both mice. Cells are colored by cluster. Schematic drawing roughly highlights how MLN and lung cells are separated on UMAP. (**B**) Stacked bar plot on the presence of expanded clones per cluster. Expansion level relates to overall presence in the dataset. (**C**) Fraction

*Figure 4 continued on next page*

*Figure 4 continued*

of cells in MLN, lung, and both organs in relation to clone expansion. Numbers above indicate proportion of expanded cells in total population. (**D**) Clonal relatedness between clusters. The stacked bar to the left gives the fraction of each cluster in the dataset as a reference (proliferating cluster was excluded). Stacked bar graphs to the right visualize for every expanded clone of a cluster where other members of a clone are found (cluster distribution). Numbers above bars represent the number of cells that each bar represents. Bars for clusters with only few expanded clones are not shown. Black horizontal lines separate the MLN and lung clusters in the bar graphs. Cluster of cells with a migratory signature is highlighted as 'bridge cluster'. (**E**) Top 5 expanded clones by occurrence in MLN (left), lung (center), or in total dataset (right).

In conclusion, expanded clones in the Th2 effector population show no evidence for preferential usage of particular TCRα or TCRβ chains or chain combinations.

## Definition of abundant CDR3 motifs in Th2 cell of Nb-infected mice

As we only found moderately expanded clones and minor signs for preferentially used TCR chains or chain combinations in our data, we performed a finer grained analysis. T-cell antigen specificity and affinity are mainly confined by variable regions of the TCR (mainly CDR3) (*Rock et al., 1994*). Therefore, we investigated if specific CDR3 motifs that might indicate potent Nb-specific effector function are repeatedly found in MLN and lung after Nb infection.

We chose the 10 most abundant CDR3 sequences on amino acid level that potentially include motifs relevant for anti-helminth immunity and highlight their abundance in different organs and mice (*Figure 5C*). TCR data of naive peripheral blood T cells served as a reference to identify germline-associated CDR3 regions. The most abundant sequence CVVGDRGSALGRLHF was found in all samples (*Figure 5C*), including the peripheral blood and represents the CDR3 motif found in the invariant TCRα chain (Vα14-Jα18) of NKT cells (*Lantz and Bendelac, 1994*). This chain is coexpressed with a variety of different TCRβ chains (not shown) and cells that express such TCRs are primarily found in the innate-like/NKT cluster (MLN6) (*Figure 1E*). As this cluster also contains some cells with expression of TCRγ chain segments in addition to functional TCRα and TCRβ chains we were able to compare if expression of the invariant TCRα is correlated to TCRγ expression. Indeed, on the one hand, 38% of cells that expressed any TCRγ constant or variable chain segment also expressed the invariant TCRα chain and on the other hand, 62% of cells with the invariant TCRα chain expressed any TCRγ constant or variable chain segment (detection of TCRγ and the invariant TCRα chain in the same cell: correlation 0.48; $p < 10^{-6}$). This might suggest a close relatedness of IL-4-expressing γδ T cells with IL-4-expressing NKT cells in a way that cells unsuccessful or not yet successful to generate a functional γδ TCR preferentially develop into αβNKT cells. Alternatively, NKT cells could induce low level of TCRγ gene expression for other, unknown reasons.

Of the remaining nine most abundant CDR3 motifs of TCRα or TCRβ chains, seven are not found in the naive peripheral blood sample (*Figure 5C*), which implies an increased probability for them to represent specificity for Nb antigens. Furthermore, only one of these motifs (CAIDPSGSWQLIF) is expressed in both analyzed organs and both mice, which implies that it could be a preferentially selected motif during Nb infection.

We next determined CDR3 motifs that are part of abundant motif combinations (*Figure 5D*, left panel). As expected, these overlap with the most expanded clones (*Figure 4E*) as cells of an expanded clone always use the same chain combination. Only one of the five TCRα CDR3 motifs (CAAEAGTG-GYKVVF) was associated with expansion in more than one clone (two clones with same TCRα CDR3 motif but different TCRβ CDR3 motifs). In addition, all five depicted TCRα CDR3 motifs present in abundant pairings are also present in unique pairings with other TCRβ CDR3 motifs. This implies that these motifs are not restricted to an exact TCRα/TCRβ combination or a single clone to be recruited to the Th2 compartment.

As others described (*He et al., 2002*; *Padovan et al., 1993*; *Padovan et al., 1995*) we find T-cell clones with expression of two TCRα/TCRβ chain-encoding genes. At least in highly abundant combinations it is unlikely that these are technical artifacts due to contamination with RNA from another cell during library preparation. The clone with the most frequently found combination of CDR3 motifs (clone MLN1) expresses one TCRβ and two TCRα chains, both on average with similar umi counts. Whether both TCRα chains are successfully translated is not known. Of the five depicted TCRα CDR3 motifs, often present in successful CDR3 combinations, four were found in some cells that expressed

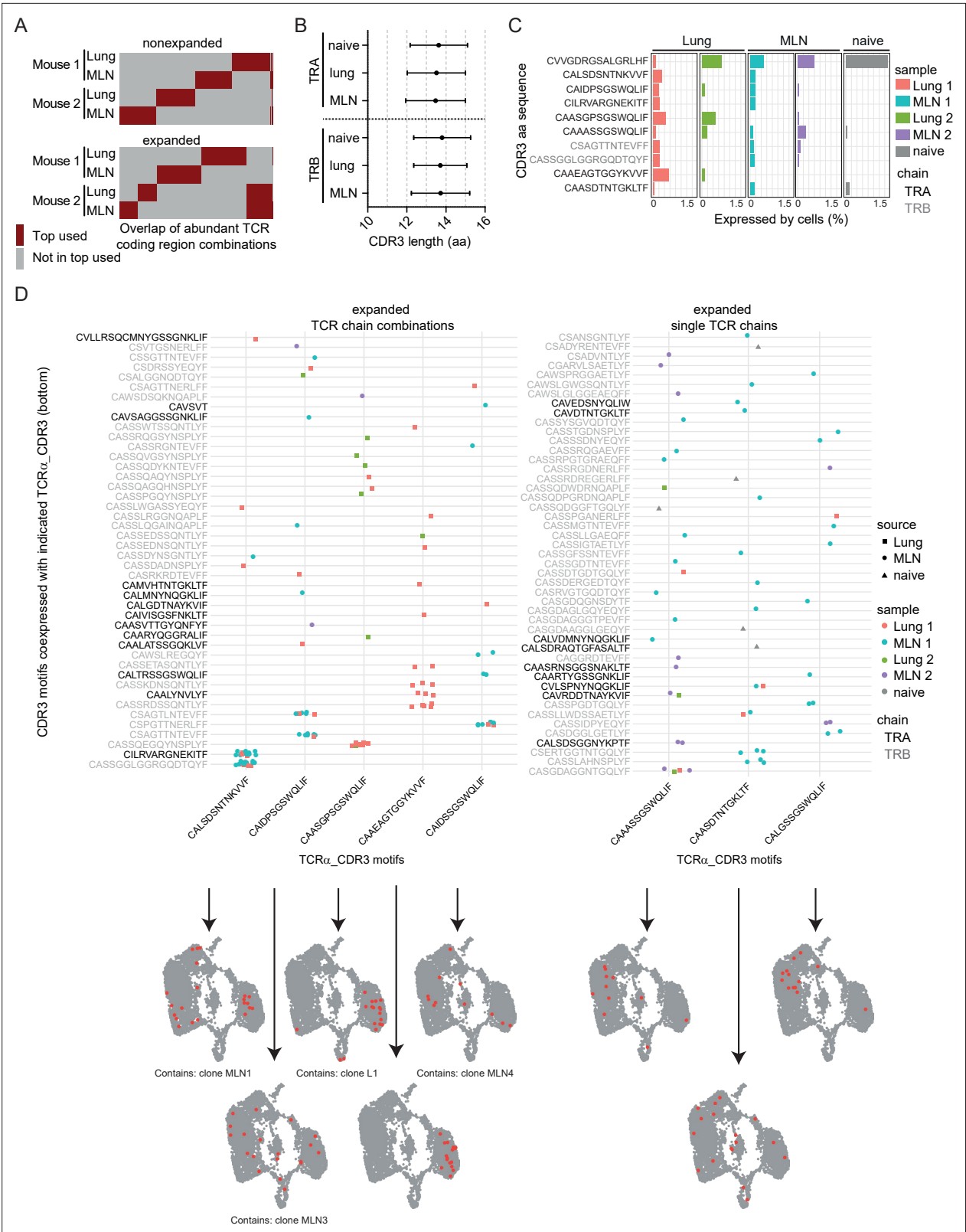

**Figure 5.** Expanded CDR3 motifs in Th2 cells of *Nippostrongylus brasiliensis* (Nb)-infected mice. T-cell receptor (TCR) repertoire analysis of mesenteric lymph node (MLN) and lung Th2 cells at day 10 post Nb infection. (**A**) TCR segment combination analysis for overlap of the hundred most commonly used TCR segment combinations (V, J, and C region for TCRα; V, D, J, and C region for TCRβ) between different mice and organs. Analysis is performed separately for nonexpanded and expanded clones. (**B**) Amino acid sequence length of TCRα and TCRβ CDR3 regions. We compare CDR3 regions

*Figure 5 continued on next page*

*Figure 5 continued*

from peripheral blood T cells of naive wild-type C57BL/6 mice (naive) with CDR3 regions of Th2 cells from MLN and lung of Nb-infected mice. (**C**) Most abundant CDR3 amino acid sequences in cells of dataset presented as percent of each sample. (**D**) Combinations of TCRα-related CDR3 motifs (*x*-axis) with indicated CDR3 motifs of TCRβ or a second TCRα chain (*y*-axis) that occurred more than once were counted. Then TCRα chains were ranked by abundance in these combinations (left). We also present highly expanded TCRα_CDR3 sequences found in combination with various TCRβ or TCRα chains (right). At the bottom, cells that express the corresponding CDR3 sequences from the *x*-axis are highlighted on top of UMAP representation of the dataset. We also indicate if a CDR3 sequence is associated with the top expanded clones (*Figure 4E*) below UMAP plots.

The online version of this article includes the following source data and figure supplement(s) for figure 5:

**Source data 1.** R script for analyzes of expanded CDR3 motifs from TCR single cell sequencing data.

**Figure supplement 1.** T-cell receptor (TCR) repertoire analysis of mesenteric lymph node (MLN) and lung Th2 cells at day 10 post *Nippostrongylus brasiliensis* (Nb) infection.

**Figure supplement 2.** CDR3 length of Th2 TCRα and TCRβ chains 10 days post *Nippostrongylus brasiliensis* (Nb) infection.

more than two TCR chains but this frequency is in the expected range for T cells (*Alam and Gascoigne, 1998*; *Balomenos et al., 1995*; *Davodeau et al., 1995*; *Dupic et al., 2019*).

In addition to CDR3 motifs found frequently in combinations (expanded clones), we also find abundant CDR3 motifs combined with various other unique TCR chains (present in several nonexpanded clones) (*Figure 5D*, right panel). These could include CDR3 motifs that provide anti-Nb specificity but failed to induce substantial expansion or accumulation of Th2 cells expressing such TCRs.

In line with a slightly preferential usage of the *Trbv1* gene in expanded compared to nonexpanded Th2 cells we find that three of the expanded TCRα CDR3 motifs (CAIDPSGSWQLIF, CAIDSSGSWQLIF, and CAASDTNTGKLTF) are preferentially coexpressed with *Trbv1*, which suggest that *Trbv1* might be relevant for the immune response against Nb.

## Functional in vivo characterization of effector-associated TCRs

As the TCR response to Nb is highly heterogeneous and as we did not find obvious dominant clones, TCR chain combinations or CDR3 motifs shared between clones, we selected TCRs of three moderately expanded clones with members in the IL-5/IL-13 expressing region of effector-associated cluster L1 (*Figure 6A*, *Figure 6—figure supplement 1A*). We hypothesized that TCRs from effector cluster-associated clones might contribute to the development of potent effector cells. The TCRs were retrovirally transduced in hematopoietic stem cells isolated from Rag1-ko mice (not able to rearrange B- and T-cell receptors) that were then transferred into irradiated recipients to generate TCR retrogenic mice (*Holst et al., 2006*). Eight weeks after transfer, we verified the presence of T cells that contain the introduced TCR α- and β-chain constructs in MLN and spleen of the retrogenic RAG1ko recipient mice (*Figure 6—figure supplement 1B*). T cells with the transgenic TCR chains (α and β) were used for transfer experiments into mice subsequently infected with Nb (*Figure 6B*). At day 9 post infection expansion of T cells with the transgenic TCRs was measured by flow cytometry as a readout for Nb reactivity. As controls, mice were either left uninfected or they were infected with another parasitic worm (*Heligmosomoides polygyrus*) to control for an unspecific expansion in response to worms. We observed a moderate but Nb-specific in vivo expansion for one of the three clones (NB-T2) highlighting that the chosen approach is suitable for the selection and reexpression of Nb-specific TCRs (*Figure 6C,D*). To our knowledge, this is the first identification of an in vivo validated Nb-specific TCR which might be used in the future to characterize the T-cell response to Nb with a given TCR specificity.

## Discussion

Th2 heterogeneity, organ crosstalk, and tissue-specific immunity are increasingly appreciated (*Schoettler et al., 2019*; *Szabo et al., 2019a*; *Szabo et al., 2019b*). Here, we applied combined transcriptome and TCR clonotype analysis on Th2 cells across organs upon Nb infection. We identified lung- and MLN-specific gene signatures as well as subpopulations with shared migration and effector/resident memory profiles between organs. We find that expression of tissue damage-associated cytokine coding genes *Il13* and *Il5* is restricted to the effector/resident memory populations in lung and MLN. Interestingly, these clusters also contain transcriptionally similar cells that express *Il10* but widely lack expression of the Treg marker-encoding gene *Foxp3*. Similar cells have been described in the skin at

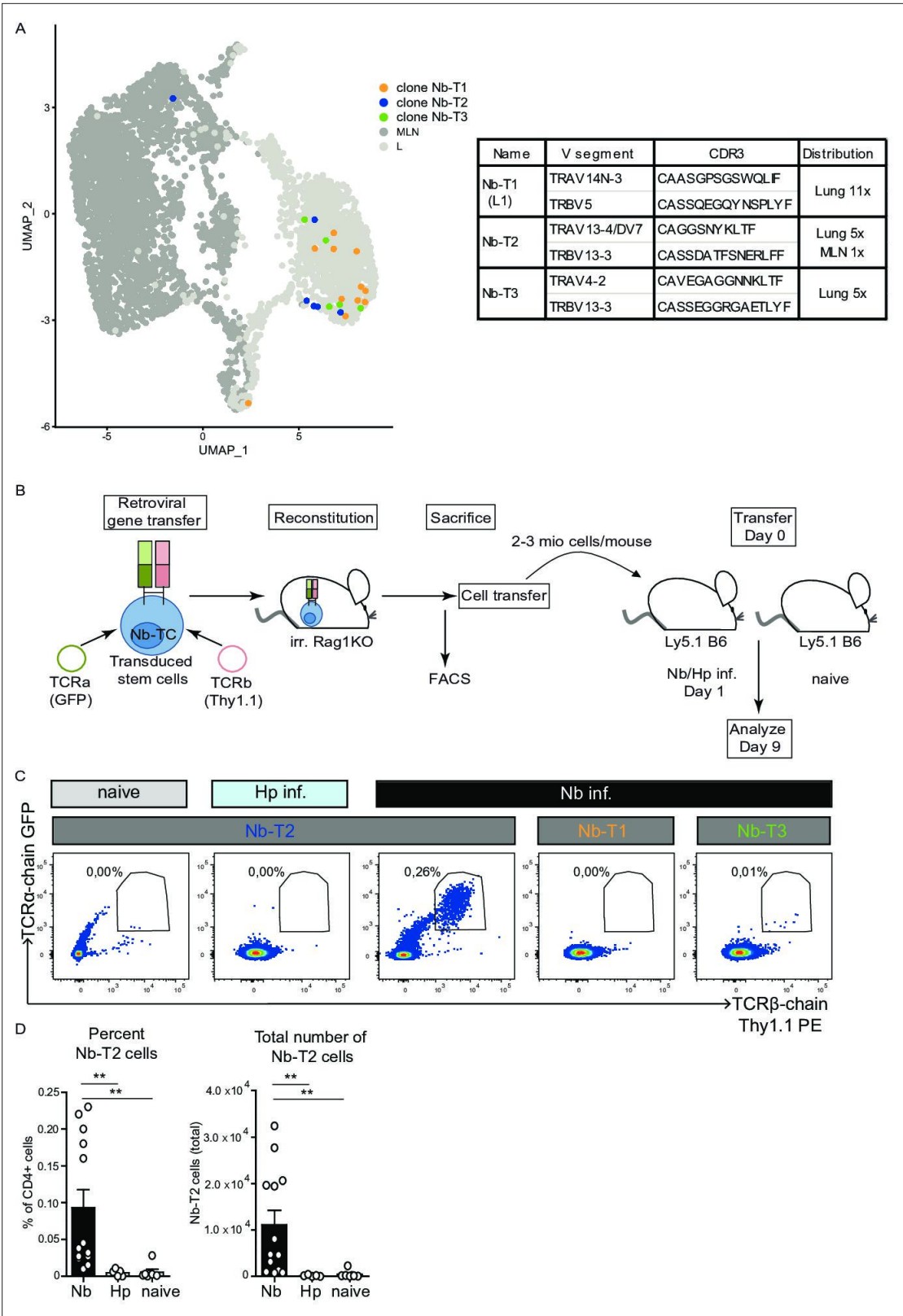

**Figure 6.** Expansion of Nb-T2 T-cell receptor (TCR) cells in mesenteric lymph node (MLN) after *Nippostrongylus brasiliensis* (Nb) infection. Hematopoietic stem cells were transduced with retroviral vectors for expression of potential Nb-specific clones (Nb-T1, Nb-T2, and Nb-T3) and used to generate retrogenic mice. (**A**) UMAP overlay shows distribution of TCRs selected for retrogenic expression. V gene segments, CDR3 sequence, and distribution between organs for selected lung-expanded TCR clones (Nb-T1, Nb-T2, and Nb-T3). (**B**) Eight weeks after reconstitution of irradiated

*Figure 6 continued on next page*

*Figure 6 continued*

Rag1KO mice with TCR chain and fluorescent marker-encoding retrovirus containing retrogenic Rag1 stem cells, T cells were harvested, transferred to Ly5.1 B6 wild-type mice which were then infected with Nb or *Heligmosomoides polygyrus* (Hp) as control. (**C**) Plots show the percentage of TCR transgenic cells (GFP$^+$, Thy1.1$^+$) among live CD4$^+$ cells from constructs Nb-T1, Nb-T2, and Nb-T3 in the MLN of recipient mice at day 9 post infection. (**D**) Percentage and total number of Nb-T2 cells in MLN. Quantification is based on six independent experiments. Statistical significance determined by Mann–Whitney *U* test; **p < 0.001.

The online version of this article includes the following figure supplement(s) for figure 6:

**Figure supplement 1.** Differentiation into CD4$^+$ T-cell receptor (TCR) transgenic cells.

the infection site after repeated *Schistosoma mansoni* cercaria infection where these cells have immunosuppressive functions (*Sanin et al., 2015*). Furthermore, effector/resident memory like cells in the MLN are not homogeneous and are found in two clusters of which one is an innate-like/NKT cluster. Interestingly, the NKT population in this cluster not only coexpresses the invariant NKT cell-associated invariant TCRα chain (Vα14-Jα18) together with a highly diverse repertoire of TCRβ chains but also transcripts for TCRγ chains, which implies shared developmental pathways of NKT and γδ T cells that both tend to express restricted receptor repertoires. The cluster also contains cells that express *Cxcr3*, encoding a typical Th1 marker. CXCR3 has been noted in a small fraction of Th2 cells (*Kim et al., 2001*) but was not associated with IL-4 producing NKT or γδ T cells before. These findings reveal a heterogeneous effector/resident memory pool in the Th2 population.

When we searched for activation signatures, we found a population of cells with an IFN response signature (MLN3) present in the MLN. Murine in vitro studies imply that IFN signaling is needed for proper Th2 differentiation (*Wensky et al., 2001*). Therefore, Th2 cells with an IFN response signature probably reflect cells that undergo priming or differentiation. RNA velocity analysis further defines them as a differentiation endpoint. Of note, the differentiation state not necessarily correlates with time in the Th2 compartment and cells could still be recently selected or undergo plasticity. Nevertheless, RNA velocity also indicates a fraction of proliferating cells that shows low expression of Th2 signature genes and that is likely in a stage where differentiation can be determined. This supports that proliferation and differentiation during Th2 development are linked as suggested before (*Proserpio et al., 2016*) rather than being independent and orthogonal processes as alternatively hypothesized (*Duffy and Hodgkin, 2012*). In addition to potential developmental paths, RNA velocity analysis suggests that development is faster and has a stricter directionality in the lung compared to the MLN, consistent with the view that the majority of Th2 cells in the MLN are less terminally differentiated.

Cells in the migratory clusters of both organs show a weaker organ-specific separation after dimensional reduction but rather form a 'bridge' on the UMAP that suggests effective exchange between organs. In line, TCR analysis of our Th2 cells revealed effective exchange of expanded clones between organs. However, the most expanded clones in the one organ were not the most expanded in the other organ. This might relate to different immunological preferences in different compartments or to the different larval stages in which Nb is present in lung and MLN. Of note, comparison of human bulk TCR repertoires between the lung and its draining lymph node also showed a higher intraorgan TCR repertoire overlap than between organs. This was interpreted to mean that the T cells originate from different precursor pools and recognize distinct antigens (*Schoettler et al., 2019*). Our data clearly refine that there is effective spread of Th2 effector T cells from the same pool of cells even across distant organs.

Another factor that likely protects from an overshooting Th2 response and could be subject to modulation by the worm is the expression of TNFR2 as impaired TNFR2 signaling leads to augmented Th2 responses (*Li et al., 2017*). We observed transcripts for TNFR2 preferentially in the lung compared to the MLN after Nb infection, which has not yet been described to our knowledge.

In summary, combined transcriptome and TCR clonotype analysis at single-cell resolution provides information about Th2 heterogeneity across organs and reveals relatedness of IL-4 producing NKT cells to γδ T cells. RNA velocity combined with knowledge from published data appears compatible with a model, in which poorly differentiated proliferating Th2 cells are at a decision point in their development, with IFN signaling being involved in diversification and differentiation of the Th2 compartment. Despite efficient exchange of expanded Th2 clones between distant organs, the most abundant clones seem to expand locally. As a proof of principle we functionally validated an Nb-specific TCR in vivo. The setup allows further functional characterization of expanded TCR clonotypes and

will help to investigate Th2 cell differentiation, plasticity, and memory formation in response to natural helminth pathogens.

# Materials and methods

## Key resources table

| Reagent type (species) or resource | Designation | Source or reference | Identifiers | Additional information |
|---|---|---|---|---|
| *Mus musculus*, C57BL/6 background, female | 4get mice | **Mohrs et al., 2001** | | IL-4eGFP reporter mouse |
| Antibody | anti-CD16/32, unconjugated, clone 2.4G2, Rat Monoclonal Antibody | BioXCell | | (1:200) |
| Antibody | anti-CD4, PerCP/Cyanine5.5, clone RM4-5, Rat Monoclonal Antibody | Invitrogen | 45-0042-82 | (1:200) |
| Antibody | anti-CD4, BUV737, clone RM4-5, Rat Monoclonal Antibody | BD | Cat#: 612844 | (1:100) |
| Antibody | anti-CXCR6 PerCP-eFluor 710, clone DANID2, Rat Monoclonal Antibody | Invitrogen | Cat#: 46-9186-82 | (1:100) |
| Antibody | anti-CD62L, eFluor 450, clone MEL-14, Rat Monoclonal Antibody | Invitrogen | Cat#: 48-0621-82 | (1:200) |
| Antibody | anti-CD127, BV711, clone SB/199, Rat Monoclonal Antibody | BD | Cat#: 565490 | (1:100) |
| Antibody | anti-Ly-6A/E (Sca-1), BV785, clone D7, Rat Monoclonal Antibody | BioLegend | Cat#: 108139 | (1:100) |
| Antibody | anti-CD3e, BV605, clone 145-2 C11, Armenian Hamster Monoclonal Antibody | BD | Cat#: 563004 | (1:100) |
| Antibody | anti-CD279 (PD-1), Super Bright 780, clone J43, Armenian Hamster Monoclonal Antibody | Invitrogen | Cat#: 78-9985-82 | (1:400) |
| Antibody | anti-CD74, Alexa Fluor 647, clone In1/CD74, Rat Monoclonal Antibody | BioLegend | Cat#: 151004 | (1:100) |
| Antibody | anti-CD44, BUV395, clone IM7, Rat Monoclonal Antibody | BD | Cat#: 740215 | (1:100) |
| Antibody | anti-CD185 (CXCR5), PE/Cyanine7, clone L138D7, Rat Monoclonal Antibody | BioLegend | Cat#: 145516 | (1:100) |
| Antibody | anti-integrin β7, PE, clone FIB27, Rat Monoclonal Antibody | BioLegend | Cat#: 121006 | (1:100) |
| Antibody | TotalSeq-C0301 is a mix of anti-mouse MHC class I and anti-mouse Ly-5 antibodies, clone M1/42 and 30-F11, Rat Monoclonal Antibodies; Hashtag 1 Antibody; Barcode Sequence ACCCACCAGTAAGAC; mouse 1 med LN | BioLegend | Cat#: 155861 | (3 µl for $6 \times 10^6$ cells) |
| Antibody | TotalSeq-C0302 is a mix of anti-mouse MHC class I and anti-mouse Ly-5 antibodies, clone M1/42 and 30-F11, Rat Monoclonal Antibodies; Hashtag 2 Antibody; Barcode Sequence GGTCGAGAGCATTCA; mouse 2 med LN | BioLegend | Cat#: 155863 | (3 µl for $6 \times 10^6$ cells) |
| Antibody | TotalSeq-C0303 is a mix of anti-mouse MHC class I and anti-mouse Ly-5 antibodies, clone M1/42 and 30-F11, Rat Monoclonal Antibodies; Hashtag 3 Antibody; Barcode Sequence CTTGCCGCATGTCAT; mouse 1 MLN | BioLegend | Cat#: 155865 | (3 µl for $6 \times 10^6$ cells) |
| Antibody | TotalSeq-C0304 is a mix of anti-mouse MHC class I and anti-mouse Ly-5 antibodies, clone M1/42 and 30-F11, Rat Monoclonal Antibodies; Hashtag 4 Antibody; Barcode Sequence AAAGCATTCTTCACG; mouse 2 MLN | BioLegend | Cat#: 155867 | (3 µl for $6 \times 10^6$ cells) |
| Other | LIVE/DEAD Fixable Lime (506) Viability Kit | Invitrogen | Cat#: L34990 | (1:1000) |
| Software, algorithm | R | **R Development Core Team, 2021** | Versions 3.5.1, 4.0.0 | |

*Continued on next page*

*Continued*

| Reagent type (species) or resource | Designation | Source or reference | Identifiers | Additional information |
|---|---|---|---|---|
| Software, algorithm | Cell Ranger | *Zheng et al., 2017* | Versions 2.1.1, 3.0.1, 6.1.2 | |
| Software, algorithm | Seurat | *Stuart et al., 2019* | Versions 3.1.1, 4.1.0 | |
| Software, algorithm | kallisto/bustools | *Bray et al., 2016* | Versions 0.46.2/0.40.0 | |
| Software, algorithm | scVelo | *Bergen et al., 2020* | Version 0.2.3 | |
| Software, algorithm | monocle | *Trapnell et al., 2014* | Version 2.16.0 | |
| Software, algorithm | CytoTRACE | *Trapnell et al., 2014* | Version 0.3.3 | |
| Software, algorithm | FlowSOM | *Van Gassen et al., 2015* | Version 1.20.0 | |
| Software, algorithm | CytoTree | *Dai et al., 2021* | Version 1.3.0 | |

## Mice and *N. brasiliensis* infection

IL-4eGFP reporter (4get) mice (*Mohrs et al., 2001*) aged 8–16 weeks were infected with third-stage larvae (L3). Larvae were washed in sterile 0.9% saline (37°C) and 500 organisms were injected subcutaneously (s.c.) into mice 10 days before analysis (if not indicated otherwise). To avoid bacterial infections mice received antibiotics-containing drinking water (2 g/l neomycin sulfate, 100 mg/l polymyxin B sulfate; Sigma-Aldrich, St. Louis, MO) for the first 5 days after infection. Mice were kept under specific pathogen-free conditions and were maintained in the Franz-Penzoldt Center in Erlangen. All experiments were performed in accordance with German animal protection law and European Union guidelines 86/809 and were approved by the Federal Government of Lower Franconia.

## Single-cell RNA and TCR sequencing

For each of the two performed single-cell sequencing experiments two IL-4eGFP reporter (4get) mice (*Mohrs et al., 2001*) at the age of 13 (MLN, lung comparison) or 10 weeks (medLN, MLN comparison) were infected with Nb. At day 10 after Nb infection lungs and lymph nodes (medLN/MLN) of IL-4eGFP reporter (4get) mice were harvested. Lungs were perfused with phosphate buffered saline, cut into small pieces and digested with the commercial 'Lung Dissociation kit' (Miltenyi, Bergisch Gladbach, GER) according to the manufacturer's instructions. Digested lungs and complete LNs were gently mashed through a 100 µm cell strainer. Next, a 40% percoll purification was applied on lung cells and erythrocytes were lysed with ACK-buffer (0.15 M $NH_4Cl$, 1 mM $KHO_3$, 0.1 mM $Na_2EDTA$). Then samples of both organs were incubated with Fc-receptor blocking antibody (anti-CD16/32, clone 2.4G2, BioXCell, West Lebanon, NH) and stained with anti-CD4-Percp-Cy5.5 antibody (clone: RM4-5). For the medLN/MLN comparison cells of each organ and each mouse were tagged with distinguishable hashtag abs (0.5 µg ab per $2 \times 10^6$ cells) added to the fluorescence antibody staining. IL-4eGFP+CD4+ cells were sorted and for each sample, 5000 cells (in case of MLN and lung comparison) or 6250 cells (in case of medLN and MLN comparison) were sorted. For MLN and lung comparison cells were subjected to 10× Chromium Single Cell 5' Solution v2 library preparation using the TCR-specific VDJ library kit according to the manufacturer's instructions (10× Genomics, Pleasanton, CA). Gene expression libraries were sequenced on an Illumina HiSeq 2500 sequencer using the recommended read lengths for 10× Chromium 5' v2 chemistry to a depth of at least 30000 reads per cell. VDJ libraries were sequenced as paired 150 bp reads to a depth of at least 30,000 reads per cell. For the medLN and MLN comparison, we followed the chromiumNextGEMSingleCell5_v2_Cell Surface-Protein guide (RevA). IL-4eGFP+CD4+ cells of different mice and organs were labeled with hashtag

antibodies and multiplexed for sequencing. The same multiplexing was performed for IL-4eGFP⁻CD4⁺ cells in parallel.

## Computational analysis

For the MLN and lung sequencing run we used 10× Genomics Cell Ranger to demultiplex sequencing reads, convert them to FASTQ format with mkfastq (Cell Ranger 2.1.1), align them to the murine genome (mm10 v3.0.0) and obtain TCR VDJ clonotypes, consensus sequences, contigs, and CDR3 regions (Cell Ranger 3.0.1). TCR-associated genes (VDJ and constant region genes for α, β, γ, and δ chains) were excluded but kept as metadata to avoid clustering by TCR genes. To be included, cells needed to be defined as such by Cell Ranger and to have >500 UMIs, >500 genes detected per cell, <7% mitochondrial reads, and a novelty >0.8 (log10 of gene number divided by log10 of UMIs). Data normalization, differential expression analysis, clustering (based on top 2000 highly variable genes), and dimensional reduction (UMAP based on top 15 principal components) were performed in Seurat (version 3.1.1) (*Stuart et al., 2019*) under R (version 3.5.1). For the medLN and MLN sequencing run Cell Ranger 6.1.2 was used for initial aggregation and alignment. Demultiplexing of hashtagged samples and integration analysis was performed in Seurat 4.1.0 under R 4.0.0. Other analysis were performed in the R and Seurat versions used for MLN and lung sequencing if not indicated differently. QC was also performed as for MLN and lung but TCR expression was no criteria. Gene set scores for each cell were calculated in Seurat as published before (*Tirosh et al., 2016*). Gene sets were taken or generated from published data: resident memory and circulating memory (*Rahimi et al., 2020*), IL-33 signature (*Morimoto et al., 2018*), proliferation signature genes for G2M- and S-phase to regress out cell cycle in scVelo (*Macosko et al., 2015*), other sets were from the 'Molecular Signatures Database' (*Subramanian et al., 2005*). TCR info was added as metadata to Seurat for combined clonotype and RNA-profile analysis. For RNA velocity, sequencing reads were aligned with kallisto/bustools (version 0.46.2/0.40.0) (*Bray et al., 2016*; *Melsted et al., 2021*) to a genome reference with unspliced and spliced RNA variants included (version GRCm39). Obtained information was used as input for scVelo (version 0.2.3) (*Bergen et al., 2020*) under python (version 3.8.5). UMAP information from Seurat was transferred to scVelo for consistency and dynamic and static velocity models with additional paga analysis were calculated with the scVelo package. Additionally a trajectory analysis with monocle (version 2.16.0) (*Trapnell et al., 2014*) was performed and developmental potential was additionally predicted with CytoTRACE (version 0.3.3) (*Gulati et al., 2020*). Usage of TCR chains and TCR chain combinations was calculated under R with custom scripts (*Figure 5—source data 1*). For TCR/CDR3 analysis, we used the Cell Ranger output and followed a recently developed workflow (according to the "CellaRepertorium" R package) with minor modifications. Contigs that missed a sanity check were excluded (needed to be productive, full length, high confidence, supported by >1 UMI, CDR3 length >5 amino acids). Similar CDR3 sequences were not combined (not assuming similar specificity for similar sequences) to maintain higher accuracy. We kept all TCR chains of T-cell clones with two TCRα/TCRβ chain-encoding genes expressed for the same reason. Data are available via GEO (GSE181342) and the 10× Genomics TCR reference dataset via the 10xGenomics website: PBMCs from C57BL/6 mice (v1, 150 × 150), Single Cell Immune Profiling Dataset by Cell Ranger 3.0.0, 10× Genomics (November 19, 2018). Flow cytometric data were pre-gated for living, CD4⁺ singlets and used as input for clustering with FlowSOM (1.20.0) (*Van Gassen et al., 2015*) using the CytoTree (1.3.0) Toolkit package (*Dai et al., 2021*) with transformMethod "autoLgcl" and scaling of data to give every channel the same weight. Data were downsampled to about $2.5 \times 10^5$ cells preserving FlowSOM defined cluster proportion for UMAP calculation (using fluorophore channel excluding eGFP IL-4 reporter and live/dead discriminator). Then data including UMAP information were exported to FCS format for further analysis.

## Generation of TCR retrogenic (Rg) mice

For analysis of Nb-specific TCR activation, TCR retrogenic mice were generated by transferring retrovirally transduced bone marrow stem cells into irradiated recipient mice as previously described (*Holst et al., 2006*). In brief, we generated TCR retrogenic mice for expression of the following TCRs: Nb-T1, Nb-T2, and Nb-T3 (*Figure 6—figure supplement 1A*). For generation of retroviral particles, using the ecotropic Phoenix E cell line and a standard Calcium Phosphate transfection protocol, the target genes were cloned into pMXpie or MSCV-Thy1.1 for expression of the TCR α- or β-chains, respectively.

Forty-eight hours before harvesting the bone marrow, donor mice (RAG1KO) were injected with sterile 5-fluorouracil solution i.p. One day before infection, harvested bone marrow cells were seeded into 24-well plates at a concentration of 2–3 × 10⁶ cells in 1 ml medium supplemented with IL-3 (20 ng/ml), IL-6 (50 ng/ml), and SCF (100 ng/ml). Retroviral infection of dividing cells was performed using a spinoculation protocol with RetroNectin and two infection steps. Recipient mice (Rag1KO) were sublethally irradiated (500 rad), followed by reconstitution with 0.5–2 × 10⁶ bone marrow stem cells i.v. Mice were provided with antibiotics-containing drinking water for 8 weeks. After reconstitution (8 weeks) the development of TCR transgenic cells was evaluated by flow cytometry of different organs and cells were further analyzed in a transfer experiment.

## Acknowledgements

The authors thank the Flow cytometry core facility of the FAU for cell sorting, Daniela Döhler and Kirstin Castiglione for technical assistance and members of the Voehringer lab for helpful discussions.

## Additional information

### Funding

| Funder | Grant reference number | Author |
|---|---|---|
| Deutsche Forschungsgemeinschaft | RTG1660 | David Voehringer |
| Deutsche Forschungsgemeinschaft | FOR2599_TP4 | David Voehringer |
| Deutsche Forschungsgemeinschaft | TRR130_TP20 | David Voehringer |

The funders had no role in study design, data collection, and interpretation, or the decision to submit the work for publication.

### Author contributions

Daniel Radtke, Conceptualization, Data curation, Software, Investigation, Visualization, Methodology, Writing - original draft, Writing - review and editing; Natalie Thuma, Conceptualization, Investigation, Visualization, Methodology, Writing - original draft, Writing - review and editing; Christine Schülein, Investigation; Philipp Kirchner, Arif B Ekici, Methodology; Kilian Schober, Conceptualization; David Voehringer, Conceptualization, Supervision, Funding acquisition, Writing - original draft, Writing - review and editing

### Author ORCIDs

Daniel Radtke http://orcid.org/0000-0003-3241-4542
David Voehringer http://orcid.org/0000-0001-6650-0639

### Ethics

All experiments were performed in accordance with German animal protection law and European Union guidelines 86/809 and were approved by the Federal Government of Lower Franconia.

### Decision letter and Author response

Decision letter https://doi.org/10.7554/eLife.74183.sa1
Author response https://doi.org/10.7554/eLife.74183.sa2

## Additional files

### Supplementary files
• Transparent reporting form

### Data availability
Single-cell RNA sequencing data are available via GEO with the ID GSE181342.

The following dataset was generated:

| Author(s) | Year | Dataset title | Dataset URL | Database and Identifier |
|---|---|---|---|---|
| Radtke D, Thuma N, Voehringer D | 2021 | Th2 single-cell heterogeneity and clonal interorgan distribution in helminth-infected mice | https://www.ncbi.nlm.nih.gov/geo/query/acc.cgi?acc=GSE181342 | NCBI Gene Expression Omnibus, GSE181342 |

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
