## [Editor Report]

CD4 Th2 effector cells contribute to immune responses to helminths and allergens. There exists little information on Th2 cell heterogeneity and clonal distribution between organs. In this manuscript, Radtke et al. investigate the transcriptional signatures of CD4 Th2 cells in the mesenteric lymph nodes and lungs during helminth infection. By using single cell RNA-sequencing including TCR clonotype analysis, the authors define distinct and overlapping transcriptional signatures and clonal relatedness between CD4 Th2 cells in two different tissues at the peak of a type 2 immune response in vivo.

---

## [Decision Letter]

**Decision letter after peer review:**

Thank you for submitting your article "Th2 single-cell heterogeneity and clonal interorgan distribution in helminth-infected mice" for consideration by *eLife*. Your article has been reviewed by 3 peer reviewers, and the evaluation has been overseen by a Reviewing Editor and Dominique Soldati-Favre as the Senior Editor. The following individuals involved in review of your submission have agreed to reveal their identity: Carolyn Genevieve King (Reviewer #1); Ashraful Haque (Reviewer #3).

Essential revisions:

Although the work has potential for publication in eLife, it requires essential additional data to support the central claims of the paper. Each reviewer raised substantive concerns (see below) that need to be resolved experimentally. For instance, new experimental approaches to be considered are: (1) perform a time course (2) extend analysis to lung draining lymph nodes and the gut itself and (3) if possible validation of some of the novel findings in mice. Finally, from a conceptual perspective, CD4 T cells assessed via IL-4-eGFP reporter mice cannot be strictly defined as "Th2": this should be addressed in the revised of manuscript.

*Reviewer #1 (Recommendations for the authors):*

– This reviewer is not an expert in Nb infection, so it is unclear if some of the findings have already been reported using other methods (e.g. flow cytometry for CXCR3, TNFR2, T-bet, Tcf1, etc). The manuscript would benefit from some biologic confirmation of the transcriptional analysis. For example, can the L2 population be detected by FACS (differential expression of CD62L, CCR7, S1PR1) and is this circulating "bridge" population modulated by treatment with FTY720?

– The authors perform clustering analysis on combined Th2 cells from lung and MLN. To better understand heterogeneity within a particular organ it would also be informative to show an analysis of lung and MLNs clustered separately and then assessing cluster similarity between the two organs.

– Could the authors show PCA alongside UMAP? Do velocity arrows look similar on PCA?

– The idea that differentiation occurs from a proliferating population is in line with the literature on many different T cell types (Th2 cells included, https://doi.org/10.1186/s13059-016-0957-5). The data shown here confirms this but doesn't really refer to previous work or competing hypotheses.

– IL-4+ cells were sorted; why are MLN2 clusters considered "Tfh2" rather than just "Tfh"?

– On page 6 it says "gene for PD-1 hardly detectable in the MLN but present in the lung". In Figure 2, it looks like Pdcd1 is all over the MLN (left) side as well and is even one of the discriminating genes for cluster MLN2 on the heatmap. This should be clarified.

– Could the authors comment on the time points chosen for analysis? For example, given the natural course of infection (parasite entering lung, coughed up and swallowed prior to a gut response) is it expected that lung immune cells at day 10 are primed earlier than those from MLN? Could this affect the interpretation of "static" velocity data which is only capable of showing a difference of a few hours between unspliced/spliced?

– The scVelo algorithm finds the end points (root, end) by comparing unspliced to spliced to find the 'stationary states of the velocity inferred transition matrix'. This happens to coincide with the most proliferating cells. Although is certainly plausible, could these cells seem stationary because they are currently "stuck" in cell cycle? In other words, is the signal used by scVelo being dominated by cell cycle genes and will velocity on samples with proliferating cells always see them as an endpoint? I'm not exactly sure how that could be tested, but maybe by regressing out cell cycle genes or removing a certain set of genes from the analysis?

*Reviewer #2 (Recommendations for the authors):*

There are four main experimental limitations of the study.

First, as outlined in the Public Review, the approach for isolating Il4-licensed, CD4^+^ T cells from the lung does not separate intravascular and intraparenchymal cells. Cluster L2 is suggested to represent recent immigrant "Th2 cells" into the lung parenchyma. But L2 may simply represent intravascular (circulating) Il4-licensed, CD4^+^ T cells rather than recent immigrants into the lung parenchyma. The only way to determine these two possibilities would be to perform intravascular staining and separately sort intravascular and intraparenchymal Il4-licensed, CD4^+^ T cells for scRNA-seq analysis.

Second, the choice to isolate Il4-licensed, CD4^+^ T cells from the mesenteric lymph nodes and lungs does not give insight into the similarities and differences between two organ systems (gut vs lung). To do so would require isolating cells from the intestines, mesenteric lymph nodes, lungs, and lung-draining lymph nodes. Identifying shared and distinct transcriptional features of Il4-licensed, CD4^+^ T cells from the gut and lungs is certainly of interest to the field, but the current approach does not clearly define the transcriptional signatures from these two organs.

Third, the authors analyze a single time point (day 10). scRNA-seq and ATAC-seq of CD4^+^ T cells has been performed in an allergy model of type 2 immunity at various time points (Tibbitt et al., Immunity, 2019). As a result, it remains unclear how the transcriptional signature of Il4-licensed, CD4^+^ T cells at different sites would change over the duration of helminth infection, especially given the helminth life cycle involves multiple infection stages. Performing scRNA-seq analysis at more than one time point would likely strength the manuscript by providing a longitudinal transcriptional signature. Alternatively, the manuscript could also be strengthened by taking key features from the single time point of scRNA-seq and performing flow cytometry validation over multiple time points.

Fourth, along the lines of the point above, the study provides no validation experiments of the initial two-mouse experiment. The study would benefit from validating key findings via another approach, such as flow cytometry, to help solidify the main conclusions of the manuscript.

*Reviewer #3 (Recommendations for the authors):*

I am grateful for the opportunity to review your manuscript. I felt the data was generated and analysed appropriately. In general, I felt there was a lack of clear direction in this manuscript. It was unclear what biological questions were being addressed, and importantly, what novel findings were made. While I do not necessarily feel a hypothesis is essential, a clearer objective is required. I outline below several options for improving the work, from broadening the study to other tissues, using additional trajectory inference tools, and conducting experimental hypothesis testing or validation.

Line 98-104: Initial comment on scRNAseq data concerns a gene that is not commonly studied in CD4^+^ T cells, TAGLN2. This is a confusing start to the Results section. Readers could be better orientated around the data by starting off with established T helper cell markers, such as transcription factors, chemokines and their associated receptors, cytokines and their cytokine receptors, integrins, etc, as well as gene signatures associated with broader cellular processes such as proliferation and metabolism.

Line 108-128: The authors present several observations from the data, but they lack coherence or sufficient detail to discern the biological point being made. In general the first section of the Results section, pertaining to Figure 1, present neither a clear research objective, nor a clear picture of the findings inferred from the data.

Lines 132-219: The authors present a lengthy transcriptomic assessment of clusters defined from an established unsupervised approach. While there may be merit in the description of many of these apparent clusters, the reader is essentially presented with a dry list of observations that do not form a coherent picture. Of most concern is that no inferences, conclusions or hypotheses are clearly articulated at the end of this section, and no biological validation is presented. I suggest that the authors choose to focus this section on the most important novel findings, and by proceeding back to the wet-lab, they might consider testing their novel findings experimentally.

Lines 224-248: Trajectory inference of cells harvested from a single timepoint, presented solely as the output from an RNA Velocity analysis, is of questionable merit. I suggest that any inferences from RNA Velocity should be tested using other trajectory inference tools, for example Slingshot, Monocle, PAGA etc, when conducted on more than one low-dimensional embedding, e.g adding to this manuscript use of single-cell Variational Inference (scVI) from the Yosef lab. Once a robust inference has been made, this should be tested experimentally in mice.

Lines 253-380: The authors present an extensive assessment of TCR clones across MLN and the lung, but while the analysis appears entirely appropriate, unfortunately no clear biological question is conveyed, and the reader is left wondering at the significance of these data. Crucially, no new hypotheses are presented, and no experimental testing undertaken. To extend this study, I suggest formulation and testing of a hypothesis based on the TCR data.

[Editors' note: further revisions were suggested prior to acceptance, as described below.]

Thank you for resubmitting your work entitled "Th2 single-cell heterogeneity and clonal distribution at distant sites in helminth-infected mice" for further consideration by *eLife*. Your revised article has been evaluated by Dominique Soldati-Favre (Senior Editor) and a Reviewing Editor.

CD4 Th2 effector cells contribute to immune responses to helminths and allergens. There exists little information on Th2 cell heterogeneity and clonal distribution between organs. In this manuscript, Radtke et al. investigate the transcriptional signatures of CD4 Th2 cells in the mesenteric lymph nodes and lungs during helminth infection. By using single-cell RNA-sequencing including TCR clonotype analysis, the authors define distinct and overlapping transcriptional signatures and clonal relatedness between CD4 Th2 cells in two different tissues at the peak of a type 2 immune response in vivo.

The authors have done a reasonable number of additional experiments and their analyses greatly improved the original study. As described below a couple of minor revisions need to be, however, addressed and uploaded in a final revised version of the manuscript.

1. The authors need to make clear that some of the differences that they have found are differences related to T cells found in peripheral tissue vs. T cells found in a lymph node as opposed to differences in T cells found in the lung vs. gut.

2. The authors also need to clarify more explicitly that since they did not distinguish intraparenchymal T cells from intravascular T cells in their lung preparations and single-cell analyses, that some of the cells included in their data set were likely intravascular (blood) cells and not intraparenchymal cells.

3. It was not clear as to how many mice were used for the repeat scRNAseq experiment.

4. Finally, the TCR cloning and retrogenic experiments that were presented constituted a strength of this revision and certainly opens the door for further interesting experiments with this infection model. Along that line, the authors missed an interesting opportunity to phenotype the Nb TCR transgenic cells that expanded in vivo. Moreover, it might be useful to specify in the Results or Discussion whether TCR transgenic reagents currently exist for this infection model. If they do not exist, the authors might present them to help researchers follow helminth-specific CD4 T cell responses.

---

## [Author Response]

Reviewer #1 (Recommendations for the authors):– This reviewer is not an expert in Nb infection, so it is unclear if some of the findings have already been reported using other methods (e.g. flow cytometry for CXCR3, TNFR2, T-bet, Tcf1, etc). The manuscript would benefit from some biologic confirmation of the transcriptional analysis. For example, can the L2 population be detected by FACS (differential expression of CD62L, CCR7, S1PR1) and is this circulating "bridge" population modulated by treatment with FTY720?

As suggested by the reviewer we validated the presence of transcriptionally defined L2 markers as part of two experiments.

In the first we stained for L2 cells as the CD4^+^CD62L+ population to address the concern that the L2 population could represent cells from the vasculature rather than cells located in the tissue (new Figure 2—figure supplement 1). We also tried to stain for S1PR1 as part of the experiment but the antibody stain didn’t work for unknown reasons.

In the second experiment we validated our transcriptionally defined clusters on protein level by flow cytometry. We find a CD62L+CXCR6- population compatible with the L2 cluster and a cluster compatible with L1. However, on protein level the clear separation into two lung populations found on the transcriptional level is less obvious.

To further biologically confirm transcriptional results we retrovirally expressed three TCRs from our study and confirm Nb-specific T cell expansion for one of them in vivo (new Figure 6). We thank the reviewer for the suggestions and hope we strengthened the manuscript with the added data.

– The authors perform clustering analysis on combined Th2 cells from lung and MLN. To better understand heterogeneity within a particular organ it would also be informative to show an analysis of lung and MLNs clustered separately and then assessing cluster similarity between the two organs.

To better understand heterogeneity and similarities between organs we split the datasets by organ and integrated the MLN and the lung data subsequently again based on shared “anchor” genes (new Figure 2—figure supplement 2). With this approach cells with the highest similarities between organs are rather forced to cluster together and don’t separate by organ anymore. Unbiased clustering of the integrated data defines similar clusters to the ones in the original non-integrated dataset, suggesting that the original clustering represents sample heterogeneity well. However, cells of effector-associated cluster L1 and MLN4 as well as the “bridge” cluster cells of L2 and MLN5 cluster together, supporting our initial description that there are similarities between effector and “bridge” clusters across organs.

– Could the authors show PCA alongside UMAP? Do velocity arrows look similar on PCA?

We provide the requested PCA data (PCA 1 to 4) with overlaid velocity arrows (new Figure 2—figure supplement 3). For all combinations of the top PCAs plotted, the arrows in the proliferating cluster indicate development towards the other clusters, compatible with a relatively undifferentiated/undecided state of the proliferating cells.

– The idea that differentiation occurs from a proliferating population is in line with the literature on many different T cell types (Th2 cells included, https://doi.org/10.1186/s13059-016-0957-5). The data shown here confirms this but doesn't really refer to previous work or competing hypotheses.

We apologize for not citing the mentioned work in the first version of the manuscript, added it to the introduction (Lines 58-61) and discuss competing hypothesis (Lines 535-537). In the Proserpio et al. manuscript they define three consecutive developmental states based on cell cycle tracking and an IL-13 reporter. The first is associated with activation but a lack of proliferation, the second is proliferation associated and the third is characterized by proliferation and IL-13/cytokine expression. Likely due to the low cell numbers in the study the model system focuses on the strongest effects observed and is less detailed compared to our analysis. As the reviewer points out the model is in general compatible with our observations.

– IL-4+ cells were sorted; why are MLN2 clusters considered "Tfh2" rather than just "Tfh"?

Tfh2 cells are considered to be CXCR3 and CXCR6 negative and to produce IL-21 and IL-

4. Due to stochastic noise in single cell data it is hard to confidently define the Tfh cells in our data as Tfh2 according to the definition. Therefore, we agree with the reviewer and rather describe the cells as IL-4 producing Tfh cells.

– On page 6 it says "gene for PD-1 hardly detectable in the MLN but present in the lung". In Figure 2, it looks like Pdcd1 is all over the MLN (left) side as well and is even one of the discriminating genes for cluster MLN2 on the heatmap. This should be clarified.

We apologize for the mistake and deleted the sentence. The sentence was meant to describe differences between the lung and MLN “bridge” clusters but ended up at the wrong position after text rearrangement.

– Could the authors comment on the time points chosen for analysis? For example, given the natural course of infection (parasite entering lung, coughed up and swallowed prior to a gut response) is it expected that lung immune cells at day 10 are primed earlier than those from MLN? Could this affect the interpretation of "static" velocity data which is only capable of showing a difference of a few hours between unspliced/spliced?

The 10x sequencing data at one time-point represents a snapshot. It was chosen as d10 typically represents the peak of the immune response in the Nb model and allows comparison of the TCR repertoire in different organs of the same mouse. By choosing Nb we apply a highly relevant, complex antigen model which increases the likelihood to define mechanisms transferrable to human hookworm infection. However, given the dynamic nature of the model one drawback is that there is no definitive time point at which a defined comparable amount of antigen reaches the distant analyzed organs. As we describe in the manuscript Nb passes several larval stages and organs during infection. However, stage 4 larvae are found in lung and gut and certainly provide a shared antigen basis between both sites (migration stage from lung to intestine; Camberis et al. Curr Protoc Immunol. 2003). The generally described time course of the Nippo infection that we refer to does further not accurately reflect antigen distribution as one will probably find a low grade of systemic distribution i.e. due to dying worms whose components are released (added to manuscript lines 97-104). Therefore, antigen likely reaches the lung and MLN in a narrow time frame. Nevertheless, the newly added flow cytometric infection time course data suggest a similar dynamic of Th2 populations between the mediastinal lymph nodes (medLN; lung draining LN) and the mesenteric lymph nodes (MLN) (new Figure 3). Furthermore, the newly generated single cell data of medLN and MLN cells suggests that cells of the distant lymph nodes cluster are quite similar and all clusters (also representing developmental stages) seem to be shared between both organs indicating that there is no later developmental stage present in the medLN that would be missed in our MLN samples (new Figure 1C,D and Figure 1—figure supplement 3). As pointed out by the reviewer the velocity data is only capable to suggest recent changes and developments in one cell and the presence of cells at different developmental stages in the data-set then allows to infer developmental directions over the whole population. As all clusters are shared between d6, d8 and d10 and also between medLN and MLN in our flow cytometric analysis and in the newly added single cell dataset after Nb infection, we rather expect a continuous development of Th2 cells. Therefore, the different developmental stages should be covered in our dataset even though we focus on a single time-point for the RNAseq analysis which hence should be suitable for an RNA velocity analysis.

– The scVelo algorithm finds the end points (root, end) by comparing unspliced to spliced to find the 'stationary states of the velocity inferred transition matrix'. This happens to coincide with the most proliferating cells. Although is certainly plausible, could these cells seem stationary because they are currently "stuck" in cell cycle? In other words, is the signal used by scVelo being dominated by cell cycle genes and will velocity on samples with proliferating cells always see them as an endpoint? I'm not exactly sure how that could be tested, but maybe by regressing out cell cycle genes or removing a certain set of genes from the analysis?

The reviewer brings up a relevant point in highlighting limitations of RNA velocity. To address the reviewers concern, we regressed out cell-cycle but kept the original UMAP coordinates. Developmental direction was similar to the attempt with cell-cycle included and inferred root- and end-points were also comparable (Figure 2—figure supplement 3E). We further used CytoTRACE (Gulati et al. Science 2020), a tool described to be robust against cell cycle effects that aims to find cells with high differentiation potential. The tool also suggests our proliferation cluster as the population with the highest differentiation potential (Figure 2—figure supplement 3F).

Reviewer #2 (Recommendations for the authors):There are four main experimental limitations of the study.First, as outlined in the Public Review, the approach for isolating Il4-licensed, CD4^+^ T cells from the lung does not separate intravascular and intraparenchymal cells. Cluster L2 is suggested to represent recent immigrant "Th2 cells" into the lung parenchyma. But L2 may simply represent intravascular (circulating) Il4-licensed, CD4^+^ T cells rather than recent immigrants into the lung parenchyma. The only way to determine these two possibilities would be to perform intravascular staining and separately sort intravascular and intraparenchymal Il4-licensed, CD4^+^ T cells for scRNA-seq analysis.

According to the reviewers suggestion we performed an intravascular staining to discriminate cells within the lungs from those in the circulation (new Figure 2—figure supplement 1). According to the vascularity staining method (with slightly increased time between i.v. and sacrifice compared to Anderson, KG et al., Nat Protoc, 2014 for higher probability of successful staining) our L2 lung cluster is a mixture of circulating cells and immigrating cells which we describe in the text (lines 210-213). The finding that the cells from the vasculature and the cells that we classified as “immigrants” seem to cluster together based on the similarity of their expression profiles on our UMAP further supports the classification of the L2 tissue fraction as “recent immigrants”. We thank the reviewer for this helpful comment which improved the manuscript.

Second, the choice to isolate Il4-licensed, CD4^+^ T cells from the mesenteric lymph nodes and lungs does not give insight into the similarities and differences between two organ systems (gut vs lung). To do so would require isolating cells from the intestines, mesenteric lymph nodes, lungs, and lung-draining lymph nodes. Identifying shared and distinct transcriptional features of Il4-licensed, CD4^+^ T cells from the gut and lungs is certainly of interest to the field, but the current approach does not clearly define the transcriptional signatures from these two organs.

As part of the revision we additionally provide newly generated single cell sequencing data that compares medLN and MLN cells at day 10 after Nb infection and find that UMAP clusters are largely overlapping between medLN and MLN (new Figure 1—figure supplement 3). This suggests that there is no broad medLN/MLN site specific signature present that would force the medLN and MLN cells to cluster apart. Addition of the newly generated medLN/MLN data on the lung/MLN UMAP based on shared anchors (Stuart et al. Cell. 2019) also leads to a clear separation between all LN and lung cells supporting that cells don’t cluster due to a site-specific respiratory tract vs intestinal tract signature but likely based on developmental stages (new Figure 1C,D). An exception are defined effector clusters that show signs of a site-specific signature (L1 expresses *Ccr8*, MLN4 and MLN6 express *Ccr9*, differences are also suggested by clustering described in lines 247-252). A similar phenotype to the one observed on the transcriptional level is observed when we cluster medLN/MLN and lung cells based on scRNAseq suggested surface marker expression after flow cytometric analysis, extending analysis to medLN on protein level (new Figure 3). It would have also been interesting to include lamina propria T cells as the reviewer suggested but we were not able to extract high quality cells at day 10 after Nb infection which is a common limitation in the Nb model.

Third, the authors analyze a single time point (day 10). scRNA-seq and ATAC-seq of CD4^+^ T cells has been performed in an allergy model of type 2 immunity at various time points (Tibbitt et al., Immunity, 2019). As a result, it remains unclear how the transcriptional signature of Il4-licensed, CD4^+^ T cells at different sites would change over the duration of helminth infection, especially given the helminth life cycle involves multiple infection stages. Performing scRNA-seq analysis at more than one time point would likely strength the manuscript by providing a longitudinal transcriptional signature. Alternatively, the manuscript could also be strengthened by taking key features from the single time point of scRNA-seq and performing flow cytometry validation over multiple time points.

As part of the revision we screened for surface marker expression in the single cell sequencing dataset on transcript level and stained these on protein level (new Figure 3 and Figure 3—figure supplement 1). This allows to follow the populations defined by scRNAseq longitudinally (d0, d6, d8, d10) by flow cytometry during Nb infection. We compared medLN, MLN and lung. The dynamic of the response in the medLN and the MLN seems similar with a small delay in the MLN compared to medLN.

Nb with its relatively well defined migratory path through the body provides a relevant complex model antigen naturally present in the respiratory tract and the intestine during infection. However, analysis of complexity and relevance does often invoke limitations. While stage 4 larvae are found in lung and gut and certainly provide a shared antigen basis between both sites (migration stage from lung to intestine; Camberis et al. Curr Protoc Immunol. 2003), we also think that there is a reasonable number of antigens shared between different larval stages and antigen (either actively secreted or from dying larvae) that are systemically distributed. However, there are probably immunogenic differences between larval stages but to analyze these is beyond the scope of the manuscript.

While i.e. Tibbitt et al. nicely define cell clusters with a limited number of cells they don’t include any TCR analysis and clonal information. Not much was known about the expansion of T cells in the different clusters in one organ and between organs and we provide relevant data in this regard. Furthermore, HDM as an allergy model might invoke different Th2 differentiation pathways as. i.e. Tfh13 cells are found in allergic settings but not in worm models (Gowthaman U, Science. 2019). With our approach on single cell level we were able to show effective distribution of a number of T cell clones in a highly heterogeneous immune response and describe and functionally validate successfully expanded clones / expanded TCR chains later on (i.e. new Figure 6). This kind of analysis has not been performed for a worm model before.

Fourth, along the lines of the point above, the study provides no validation experiments of the initial two-mouse experiment. The study would benefit from validating key findings via another approach, such as flow cytometry, to help solidify the main conclusions of the manuscript.

As noted above, we screened for surface marker expression in the single cell sequencing dataset on transcript level and measured these on protein level by flow cytometry as the reviewer suggested. This allows to follow the populations defined by scRNAseq longitudinally (d0, d6, d8, d10) during Nb infection (new Figure 3). Furthermore, we added a newly generated set of scRNAseq data which confirms and extends findings made in the initial sequencing experiment (Figure 1C,D, Figure 1—figure supplement 3 ). We also included validation experiments based on the performed TCR analysis and retrovirally expressed three TCRs from our study and confirm Nb specific expansion for one of them in vivo (new Figure 6 and Figure 6—figure supplement 1).

Reviewer #3 (Recommendations for the authors):I am grateful for the opportunity to review your manuscript. I felt the data was generated and analysed appropriately. In general, I felt there was a lack of clear direction in this manuscript. It was unclear what biological questions were being addressed, and importantly, what novel findings were made. While I do not necessarily feel a hypothesis is essential, a clearer objective is required. I outline below several options for improving the work, from broadening the study to other tissues, using additional trajectory inference tools, and conducting experimental hypothesis testing or validation.Line 98-104: Initial comment on scRNAseq data concerns a gene that is not commonly studied in CD4^+^ T cells, TAGLN2. This is a confusing start to the Results section. Readers could be better orientated around the data by starting off with established T helper cell markers, such as transcription factors, chemokines and their associated receptors, cytokines and their cytokine receptors, integrins, etc, as well as gene signatures associated with broader cellular processes such as proliferation and metabolism.

We rearranged Figure 1 and Figure 2 to improve the flow of the manuscript according to the reviewer’s suggestions.

Line 108-128: The authors present several observations from the data, but they lack coherence or sufficient detail to discern the biological point being made. In general the first section of the Results section, pertaining to Figure 1, present neither a clear research objective, nor a clear picture of the findings inferred from the data.

The part has been rearranged and set in context to the used infection model to provide a comprehensive story (lines 177-194).

Lines 132-219: The authors present a lengthy transcriptomic assessment of clusters defined from an established unsupervised approach. While there may be merit in the description of many of these apparent clusters, the reader is essentially presented with a dry list of observations that do not form a coherent picture. Of most concern is that no inferences, conclusions or hypotheses are clearly articulated at the end of this section, and no biological validation is presented. I suggest that the authors choose to focus this section on the most important novel findings, and by proceeding back to the wet-lab, they might consider testing their novel findings experimentally.

The Th2 compartment after Nb infection has not been transcriptionally analyzed in the resolution we provide with our dataset (allowed by use of sorted cells from 4get mice). Therefore, we feel that we are able to add valuable aspects by describing observed clusters. We tried to improve analysis coherency by including summary sentences (lines 127, 204-206) and validated stainability of defined surface receptor coding genes on protein level (new Figure 3). We also state that cluster definitions will be used to infer potential function of clones and their TCRs later on (lines 272-273) as we selected and functionally validated TCRs preferentially found in effector cluster L1 (new Figure 6).

Lines 224-248: Trajectory inference of cells harvested from a single timepoint, presented solely as the output from an RNA Velocity analysis, is of questionable merit. I suggest that any inferences from RNA Velocity should be tested using other trajectory inference tools, for example Slingshot, Monocle, PAGA etc, when conducted on more than one low-dimensional embedding, e.g adding to this manuscript use of single-cell Variational Inference (scVI) from the Yosef lab. Once a robust inference has been made, this should be tested experimentally in mice.

We agree that validation of results of a single computational tool is preferable. Therefore we support the original analysis by performing additional analysis, including static velocity (in comparison to the dynamic model used in the paper), paga, monocle and Cytotrace (Gulati et al. 2020) analysis as well as performing RNA velocity on PCA level as suggested by reviewer 1. We further validate transcriptional surface expression signatures on protein level by flow cytometry and use the newly established surface stain to perform time-course experiments to complement RNA velocity with experimental longitudinal data.

Lines 253-380: The authors present an extensive assessment of TCR clones across MLN and the lung, but while the analysis appears entirely appropriate, unfortunately no clear biological question is conveyed, and the reader is left wondering at the significance of these data. Crucially, no new hypotheses are presented, and no experimental testing undertaken. To extend this study, I suggest formulation and testing of a hypothesis based on the TCR data.

As suggested by the reviewer we now used the TCR analysis to select TCR candidates for subsequent functional analysis of Nb-specific TCRs (new Figure 6).

[Editors' note: further revisions were suggested prior to acceptance, as described below.]

CD4 Th2 effector cells contribute to immune responses to helminths and allergens. There exists little information on Th2 cell heterogeneity and clonal distribution between organs. In this manuscript, Radtke et al. investigate the transcriptional signatures of CD4 Th2 cells in the mesenteric lymph nodes and lungs during helminth infection. By using single-cell RNA-sequencing including TCR clonotype analysis, the authors define distinct and overlapping transcriptional signatures and clonal relatedness between CD4 Th2 cells in two different tissues at the peak of a type 2 immune response in vivo.The authors have done a reasonable number of additional experiments and their analyses greatly improved the original study. As described below a couple of minor revisions need to be, however, addressed and uploaded in a final revised version of the manuscript.1. The authors need to make clear that some of the differences that they have found are differences related to T cells found in peripheral tissue vs. T cells found in a lymph node as opposed to differences in T cells found in the lung vs. gut.

We added a corresponding statement in the Results section.

“This indicates that most of the differences are due to the comparison of a secondary lymphoid organ with a peripheral organ, and only minor differences are related to the comparison of distant sites (medLN vs MLN).” (lines 119-121)

2. The authors also need to clarify more explicitly that since they did not distinguish intraparenchymal T cells from intravascular T cells in their lung preparations and single-cell analyses, that some of the cells included in their data set were likely intravascular (blood) cells and not intraparenchymal cells.

We now highlight the aspect at the beginning of the Results section in brackets:

“…lung (also containing a remaining fraction of intravascular cells) and MLN…” (line 90)

3. It was not clear as to how many mice were used for the repeat scRNAseq experiment.

This info was included in the Materials and methods section (lines 590-592) and in the legend of Figure 1 (lines 951-952).

4. Finally, the TCR cloning and retrogenic experiments that were presented constituted a strength of this revision and certainly opens the door for further interesting experiments with this infection model. Along that line, the authors missed an interesting opportunity to phenotype the Nb TCR transgenic cells that expanded in vivo. Moreover, it might be useful to specify in the Results or Discussion whether TCR transgenic reagents currently exist for this infection model. If they do not exist, the authors might present them to help researchers follow helminth-specific CD4 T cell responses.

We agree that it would be interesting to further generate and characterize Nb-specific TCRtg mice. However, this goes beyond the scope of this manuscript and should be performed in detail elsewhere. We added a corresponding statement to report that no TCR transgenic reagents currently exist for Nb infection model: “To our knowledge this is the first identification of an in vivo validated Nb-specific TCR which might be used in the future to characterize the T cell response to Nb with a given TCR specificity.” (lines 505 and 507) The generation of the retrogenic cells is described in the Materials and methods section (lines 487-505) with additional info in Figure 6—figure supplement 1.